# Leveraging Locality and Robustness to Achieve Massively Scalable Gaussian Process Regression

**Robert Allison**
Department of Mathematics
Bristol University
marfa@bristol.ac.uk

**Anthony Stephenson**
Department of Mathematics
Bristol University

**Samuel F**
Alan Turing Institute

**Edward Pyzer-Knapp**
IBM Research

## Abstract

The accurate predictions and principled uncertainty measures provided by GP regression incur $\mathcal{O}(n^3)$ cost which is prohibitive for modern-day large-scale applications. This has motivated extensive work on computationally efficient approximations. We introduce a new perspective by exploring robustness properties and limiting behaviour of GP nearest neighbour (GPnn) prediction. We demonstrate through theory and simulation that as the data-size $n$ increases, accuracy of estimated parameters and GP model assumptions become increasingly irrelevant to GPnn predictive accuracy. Consequently, it is sufficient to spend small amounts of work on parameter estimation in order to achieve high MSE accuracy, even in the presence of gross misspecification. In contrast, as $n \to \infty$, uncertainty calibration and NLL are shown to remain sensitive to just one parameter, the additive noise-variance; but we show that this source of inaccuracy can be corrected for, thereby achieving both well-calibrated uncertainty measures and accurate predictions at remarkably low computational cost. We exhibit a very simple GPnn regression algorithm with stand-out performance compared to other state-of-the-art GP approximations as measured on large UCI datasets. It operates at a small fraction of those other methods' training costs, for example on a basic laptop taking about 30 seconds to train on a dataset of size $n = 1.6 \times 10^6$.

## 1 Introduction

We first briefly review the computational cost of exact GP regression and the motivation for this paper: Given $n$ training samples $X, \boldsymbol{y}$, where $X \in \mathbb{R}^{n \times d}$ has feature vector $\boldsymbol{x}_i \in \mathbb{R}^d$ in its $i$'th row and $\boldsymbol{y} \in \mathbb{R}^n$, exact GP regression [36] makes use of an $n \times n$ gram matrix $K = K_{X,\boldsymbol{\theta}}$ constructed from a pre-specified positive definite covariance function $c(\cdot, \cdot) : \mathbb{R}^d \times \mathbb{R}^d \to \mathbb{R}_+$ together with length-scale, additive-noise variance and kernel-scale "hyperparameters" $\boldsymbol{\theta} = (l, \sigma_\xi^2, \sigma_f^2)$. In the training phase estimates of the hyperparameters, $\hat{\boldsymbol{\theta}} = (\hat{l}, \hat{\sigma}_\xi^2, \hat{\sigma}_f^2)$, are obtained by minimising the loss function

$$\text{loss}(\boldsymbol{\theta}) = -\log p(\boldsymbol{y}|X, \boldsymbol{\theta}) = \frac{1}{2}\{\boldsymbol{y}^T K_{\boldsymbol{\theta}}^{-1} \boldsymbol{y} + \log |K_{\boldsymbol{\theta}}| + n \log(2\pi)\}. \tag{1}$$

Then for subsequent predictions the predictive distribution at a point $\boldsymbol{x}^* \in \mathbb{R}^d$ is defined by

$$y^* \mid X, \boldsymbol{y} \sim \mathcal{N}(\mu^*, \sigma^{*2}) \tag{2}$$

$$\mu^* = \boldsymbol{k}^{*T} K^{-1} \boldsymbol{y} \tag{3}$$

$$\sigma^{*2} = \hat{\sigma}_f^2 - \boldsymbol{k}^{*T} K^{-1} \boldsymbol{k}^* + \hat{\sigma}_\xi^2 \tag{4}$$

37th Conference on Neural Information Processing Systems (NeurIPS 2023).

where $K = K_{\hat{\theta}}$ with components $[K]_{ij} = k_{\hat{\theta}}(\boldsymbol{x}_i, \boldsymbol{x}_j)$; the vector $\boldsymbol{k}^*$ has components $k_i^* = k_{\hat{\theta}}(\boldsymbol{x}_i, \boldsymbol{x}^*)$, and $k_{\boldsymbol{\theta}}(\boldsymbol{x}, \boldsymbol{x}') = \sigma_f^2 c(\boldsymbol{x}/l, \boldsymbol{x}'/l) + \delta_{\boldsymbol{x}, \boldsymbol{x}'} \sigma_\xi^2$ with a "normalised" covariance function $c(\cdot, \cdot)$ such that $c(\boldsymbol{x}, \boldsymbol{x}) = 1$. The derivation of these steps is based on the assumption that the underlying random field is Gaussian, as is the additive noise with variance $\sigma_\xi^2$.

The single cost-$\mathcal{O}(n^3)$ step of inverting $K$ is needed repeatedly to compute the loss. Sophisticated implementations reduce this toward $\mathcal{O}(n^2)$ ([35]), but even that cost is generally impractical for $n > 10^6$. For a survey of numerous GP approximations and their reduced costs see [19].

Machine learning methods must tackle massive data problems to handle many modern day applications. Revolutionary developments in neural network methodologies achieve this, but Bayesian predictive methodologies, in particular GP regression with its major advantages of robustness and uncertainty measures, are somewhat behind the curve. This motivates development of fast implementations retaining the accuracy and well-principled uncertainty of exact GPs.

## 2 Background and Paper Outline

A feature common to all mainstream GP approximations is that training and prediction processes make joint use of the same underlying mathematical constructions. In the "subset-of-data" method the *same* subset of data is used both for parameter estimation and prediction. Similarly, in the various Bayesian committee methods ([18]) hyperparameters are estimated using a collection of subsets of data and then the *same* subsets are used in combination to make predictions. In the variational ([14, 31]) and other inducing point methods parameters are estimated using a low rank approximation to the kernel gram matrix and then the *same* low-rank matrix approximation is used to make predictions. Despite being almost universally adopted there is no obvious reason why constraining algorithms to use the same constructions for estimation and prediction will help rather than hinder the end goal of high performance at low cost. Whilst there are some passing mentions of decoupling prediction and estimation in the literature - e.g. [28, 1, 3] - it has not been adopted as a mainstream approach.

Our first observation is that allowing parameter-estimation and prediction processes to become decoupled may provide the flexibility to greatly improve cost-accuracy trade-off. As shown in Figure 1, GP approximations first obtain a point estimate of the kernel hyperparameters $\hat{\boldsymbol{\theta}}$ from training data and then feed $\hat{\boldsymbol{\theta}}$ into a predictive process. Our end-goal is only to obtain accurate and well-calibrated *predictive distributions* of $y^*$ at each target point $\boldsymbol{x}^*$; obtaining accurate parameter estimates is *not* a goal in itself. It follows that the computational budget devoted to parameter estimation need only be sufficient to provide parameters capable of delivering accurate and well-calibrated predictions.

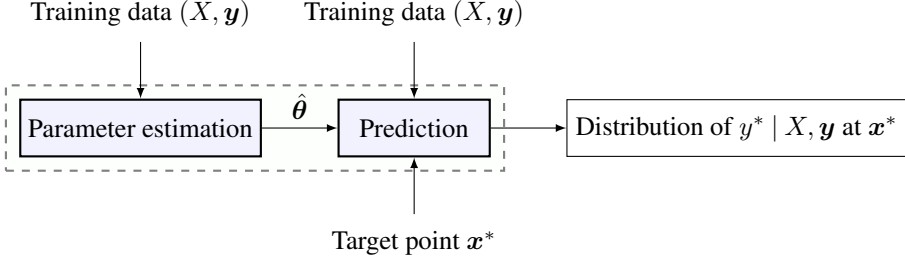

Figure 1: Flowchart of the GP regression procedure. The dashed box indicates the usual approach of combining the parameter estimation and prediction tasks under one strategy.

This need not mean that $\hat{\boldsymbol{\theta}}$ is an accurate estimate of the parameters which leads us to the second observation and main theoretical component of this paper: In section 5, theory and simulations reveal that under widely applicable circumstances, as $n$ increases the mean squared error (MSE) predictive accuracy obtained from GP nearest neighbour prediction becomes increasingly insensitive to model misspecification, i.e. insensitive to the wrong choice of covariance function, to the choice of $\hat{l}$, $\hat{\sigma}_f^2$ and $\hat{\sigma}_\xi^2$ and even insensitive to departures from Gaussian model assumptions made for the underlying stochastic process and additive noise. Similarly the negative log likelihood (NLL) predictive-accuracy becomes insensitive to all of those factors *apart from* the variance of the additive noise $\hat{\sigma}_\xi^2$. In 6.1

we describe a simple calibration step that corrects for the latter inaccuracy thereby achieving near optimal limiting NLL values in addition to well-calibrated uncertainty measures whilst leaving the well-behaved MSE values completely unaltered. We apply these overall observations to construct a highly efficient, accurate and well calibrated regression algorithm in section 6.

**Our key contributions:** Demonstration of GPnn robustness against model and parameter misspecification through theory and simulation (5); derivation of explicit formulae for the limiting MSE, NLL and calibration performance of GPnn as $n \to \infty$ (5.1); translation of this theory into a new GP approximation framework with stand-out performance relative to other state-of-the-art GP approximations (6,7.1); a simple generic method for re-calibrating uncertainty measures in GP regression with immediate applications to improving calibration of other GP approximations such as SVGP (6.1); achievement of massive scalability for GPs, for example a $100\times$ speed-up over state-of-the-art methods on a $1.6 \times 10^6$ training set whilst also improving upon their performance (7.1); demonstrating that provably best possible MSE, NLL and calibration performance can be closely approached on data that is grossly misspecified relative to GP model assumptions (7.2).

## 3   Performance Measures, Weak and Strong Calibration

Along with many other GP publications we use mean squared error MSE (or its square root RMSE) and negative log likelihood (NLL), both computed from held-aside test data, to assess predictive performance. These are simply the mean values of $e_i^* = (y_i^* - \mu_i^*)^2$ and $l_i^* = 0.5 \cdot (\log \sigma_i^{*\,2} + (y_i^* - \mu_i^*)^2/\sigma_i^{*\,2} + \log 2\pi)$ respectively. However, we find those measures alone inadequate for determining how *well calibrated* a predictive distribution is. We define "weakly calibrated" prediction to mean that $\mathrm{E}_{\mathrm{y}^* \mid X, \mathbf{y}} \left\{ (y^* - \mu^*)^2 / \sigma^{*\,2} \right\} = 1$ and accordingly use "calibration" to be a measure of how well the average value of $z_i^* = (y_i^* - \mu_i^*)^2 / \sigma_i^{*\,2}$ over test-data agrees with 1. This choice of metric (also made use of in [16]) can be motivated as follows: For a well-calibrated GP the expected squared deviation of $y_i^*$ from $\mu_i^*$ should match the corresponding predictive variance $\sigma_i^{*\,2}$ (2), i.e. $\mathrm{E}_{\mathrm{y}^* \mid X, \mathbf{y}} \{z^*\} = 1$. Hence, observing an average of $z^*$ values close to 1 is consistent with a *necessary* condition for effective calibration. In practice, we find that GP approximation methods can fall well short of this condition (e.g. see the LHS plot in Figure 4, and Table 3 for those results in tabular form) whilst this is not evident from their MSE and NLL values alone. A better measure of calibration ("strong-calibration") would have been to see how well percentiles of the predictive distributions agree with those observed in test data, e.g. see [24], but we defer such a refinement to future work.

## 4   Prediction Method and Sources of Misspecification

### 4.1   GP Nearest Neighbour Prediction

We now describe what we mean by "GP nearest neighbour (GPnn) prediction". Assume that we are given parameters $\hat{\boldsymbol{\theta}} = (\hat{l}, \hat{\sigma}_\xi^2, \hat{\sigma}_f^2)$ obtained from the parameter estimation phase of Figure 1. Then to compute the estimated pointwise-distribution of $y^*$ at $\boldsymbol{x}^*$ indicated in Figure 1 we find the $m$ nearest training-set neighbours $N = N(\boldsymbol{x}^*)$ to $\boldsymbol{x}^*$ and apply exactly the same GP prediction formulae as in (2), (3) and (4) but with $X \in \mathbb{R}^{n \times d}$ replaced by $N \in \mathbb{R}^{m \times d}$ and $\boldsymbol{y} \in \mathbb{R}^n$ replaced by $\boldsymbol{y}_N \in \mathbb{R}^m$. Note that in this setup conditioning on $N(\boldsymbol{x}^*)$ is equivalent to conditioning on the full input matrix $X$. We obtain:

$$y^* \mid N(\boldsymbol{x}^*), \boldsymbol{y}_N \sim \mathcal{N}(\mu_N^*, \sigma_N^{*\,2}) \tag{5}$$

$$\mu_N^* = \hat{\boldsymbol{k}}_N^{*T} \hat{K}_N^{-1} \boldsymbol{y}_N \tag{6}$$

$$\sigma_N^{*\,2} = \hat{\sigma}_f^2 - \hat{\boldsymbol{k}}_N^{*T} \hat{K}_N^{-1} \hat{\boldsymbol{k}}_N^* + \hat{\sigma}_\xi^2 \tag{7}$$

where we have used hatted notation in a generic manner to cover all the potential sources of misspecification (4.2) that might arise when we carry out these predictions. The $\mu_N^*, \sigma_N^{*\,2}$ parameters are substituted for $\mu^*, \sigma^{*\,2}$ when computing the performance measures described in section 3.

**Note:** In this paper we use Euclidean distance for nearest neighbour assignment but more generally could employ a metric defined by the covariance function - see A. For the covariance functions used in this paper these metrics are "equivalent" because one is a monotonic function of the other.

In our algorithmic implementations we replace an exact $m$ nearest neighbour algorithm with a much more efficient *approximate* nearest neighbour algorithm as discussed in section 6. However for the purpose of the theoretical analysis of robustness in section 5 this distinction can be ignored.

## 4.2   Sources of Misspecification

For the remainder of the paper we extend our theory and notation to encompass several (possibly simultaneous) sources of misspecification: standard GP theory assumes that data comes from a latent Gaussian random field $\mathcal{GRF}[\sigma_f^2 c(./l, ./l)]$ specified by covariance function $c(\cdot, \cdot)$ and parameters $l, \sigma_f^2$. The construction of the matrix $K_{\boldsymbol{\theta}}$ in section 1 assumes data to have arisen from this $\mathcal{GRF}$ with i.i.d $\mathcal{N}(0, \sigma_\xi^2)$ additive noise. Henceforth, we limit covariance functions $c(\boldsymbol{x}, \boldsymbol{x}')$ to be stationary, i.e. to vary only with $(\boldsymbol{x} - \boldsymbol{x}')$. The forms of (possibly simultaneous) misspecifications to be accounted for in the theoretical treatment of 5.1 are: (a) parameter $\sigma_\xi^2$ wrongly specified as $\hat{\sigma}_\xi^2$, (b) (normalised) covariance function $c(\cdot, \cdot)$ wrongly specified as $\hat{c}(\cdot, \cdot)$, (c) parameters $l, \sigma_f^2$ misspecified as $\hat{l}, \hat{\sigma}_f^2$ (relevant only if $c(\cdot, \cdot)$ *not* misspecified), (d) true additive noise is *not* Gaussian and (e) the data is generated by a non-Gaussian weakly stationary random field $\mathcal{WSRF}$ rather than a $\mathcal{GRF}$.

# 5   GP nearest neighbour Limits and Robustness

In this section we investigate the behaviour of MSE, NLL and calibration for GPnn prediction as $n \to \infty$, showing how all of these performance measures become increasingly less sensitive to hyperparameter accuracy, kernel choice and the above departures from the GP model assumptions.

## 5.1   Theory

**Assumptions:** The true generative model from which the data arises is $y_i = f(\boldsymbol{x}_i) + \xi_i$ with $\xi_i \overset{\text{iid}}{\sim} P_\xi$, $f(\boldsymbol{x}) \sim \mathcal{WSRF}[\sigma_f^2 c(./l, ./l))]$ and $y_i \mid f(\boldsymbol{x}_i) \sim P_\xi$ where the variance of the distribution $P_\xi$ is $\sigma_\xi^2$. Neither the $\mathcal{WSRF}$ nor the additive noise distribution $P_\xi$ need be Gaussian. The training $\boldsymbol{x}$ values are i.i.d. The MSE, NLL and calibration statistics on the test set are derived according to the nearest neighbour GP prediction process (4.1) and subject to any/all forms of misspecification in 4.2. Additionally, we assume that if and only if the $m^{th}$ nearest neighbour converges to the test point under $c$, then it also converges under $\hat{c}$ (A:Definition 10).

**Result:** Given a size-$n$ training set $X$ and test point $\boldsymbol{x}^*$ in the support (A:Definition 9) of the measure of $\boldsymbol{x}$, let $f_n^{\text{MSE}}(\hat{\boldsymbol{\theta}}) = \mathrm{E}_{\mathbf{y}, y^*} \{e_N^*\}$, $f_n^{\text{NLL}}(\hat{\boldsymbol{\theta}}) = \mathrm{E}_{\mathbf{y}, y^*} \{l_N^*\}$, $f^{\text{CAL}}(\hat{\boldsymbol{\theta}}) = \mathrm{E}_{\mathbf{y}, y^*} \{z_N^*\}$; where expectations are w.r.t. the true generative process for $\boldsymbol{y}$ and $y^*$ and the performance measures $e_N^*, l_N^*, z_N^*$ (section 3) are for the nearest neighbour prediction process. Note that these are the expected (rather than mean) values of the performance measures described in section 3 and the dependence on $n$ is implicit in the construction of the nearest neighbour sets $N = N_m(\boldsymbol{x}^*)$ used for prediction. Then we have:

**Theorem 1** (GPnn limits). *As $n \to \infty$, $f_n^{\text{MSE}}, f_n^{\text{CAL}}, f_n^{\text{NLL}} \to f_\infty^{\text{MSE}}, f_\infty^{\text{CAL}}, f_\infty^{\text{NLL}}$ a.e w.r.t. the (i.i.d.) measure on $\mathbf{x} \in X$ and $\mathbf{x}^*$, and pointwise as functions of $\hat{\boldsymbol{\theta}}$ where:*

*(i)* $f_\infty^{\text{MSE}}(\hat{\boldsymbol{\theta}}) = \sigma_\xi^2(1 + m^{-1}) \pm \mathcal{O}(m^{-2})$

*(ii)* $f_\infty^{\text{CAL}}(\hat{\boldsymbol{\theta}}) = \frac{\sigma_\xi^2}{\hat{\sigma}_\xi^2} \pm \mathcal{O}(m^{-2})$

*(iii)* $f_\infty^{\text{NLL}}(\hat{\boldsymbol{\theta}}) = \frac{1}{2} \left\{ \log \left( \hat{\sigma}_\xi^2(1 + m^{-1}) \right) + \frac{\sigma_\xi^2}{\hat{\sigma}_\xi^2} + \log 2\pi \right\} \pm \mathcal{O}(m^{-2})$.

*Setting $\hat{\boldsymbol{\theta}} = \boldsymbol{\theta}$ (and in particular $\hat{\sigma}_\xi^2 = \sigma_\xi^2$) in the above provides matched-parameter limiting results.*

*Proof sketch.* It is quite straightfoward to derive expressions for each of the expectations $f_n^{\text{MSE}}, f_n^{\text{CAL}}, f_n^{\text{NLL}}$ since these only depend on the known marginal covariance matrices of the (misspecified) $\mathcal{WSRF}$. We then use results concerning asymptotic convergence of Euclidean nearest neighbours, in combination with some standard linear algebra results and continuity properties, to

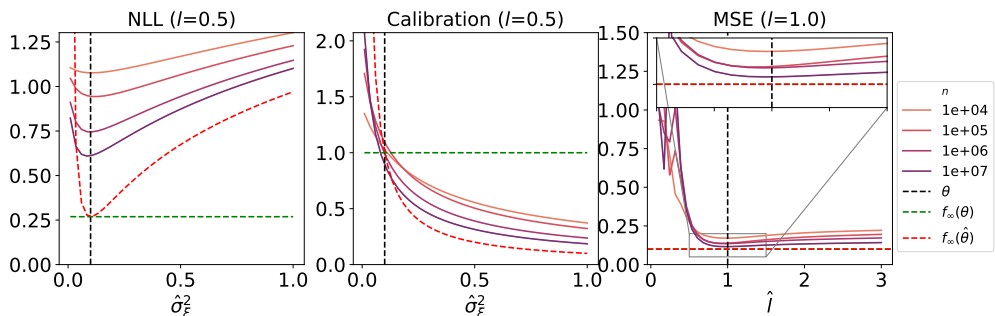

Figure 2: Behaviour of performance metrics as functions of kernel hyperparameters for increasing training set sizes $n$. The black dashed line denotes the true parameter value; the red dashed line shows the limiting behaviour as $n \to \infty$ and the green dashed line shows the limiting behaviour when the hyperparameters are correct (the red and green dashed lines coincide for MSE). True $l$ is shown in the title; additionally $\sigma_\xi^2 = 0.1$, $\sigma_f^2 = 0.9$, $d = 20$. When not varied, the assumed parameters are $\hat{\sigma}_\xi^2 = 0.2$, $\hat{\sigma}_f^2 = 0.8$, $\hat{l} = l$. Finally we generate the input data from the measure $P_{\mathbf{x}} = \mathcal{N}\left(0, \frac{1}{d}I_d\right)$.

obtain the stated limits. Note that in the expression for $f_\infty^{\mathrm{MSE}}(\hat{\boldsymbol{\theta}})$ above the right hand side must always exceed $\sigma_\xi^2$ since this is an absolute lower bound on MSE performance; likewise $f_\infty^{\mathrm{CAL}}(\hat{\boldsymbol{\theta}})$ is constrained to be non-negative. See *full proof (A)*.. $\qquad\square$

**Interpretation:** The MSE results of Theorem 1 show that to within a small factor (e.g. $m^{-1} = 0.0025$ when $m = 400$ as for all reported runs of our algorithm) the *best possible* MSE will be achieved in the limit. The NLL results also tell us (by setting $\sigma_\xi^2 = \hat{\sigma}_\xi^2$) what the best possible limiting NLL value is, but only according to the *possibly misspecified* Gaussian model. The corrupting influence of an incorrect value of $\hat{\sigma}_\xi^2$ on the limiting NLL value is clearly evident from the expression for $f_\infty^{\mathrm{NLL}}$ and the picture is similar for calibration.

**Remark 2.** *Theorem 1 shows that isotropic (e.g. RBF and Matérn) kernels converge to the best possible MSE as $n \to \infty$ even on data generated with independent lengthscales on each $\boldsymbol{x}$ coordinate.*

Note that 1 refers to pointwise convergence whereas we believe uniform convergence results should also be obtainable, e.g. perhaps of the form (or similar):

**Conjecture 3.** $\mathrm{E}_{X,\mathbf{x}^*}\left\{f_n^{\mathrm{MSE}}(\hat{\boldsymbol{\theta}})\right\} \to f_\infty^{\mathrm{MSE}} = \sigma_\xi^2(1 + m^{-1}) \pm \mathcal{O}\left(m^{-2}\right)$ *uniformly as a function of* $\hat{\boldsymbol{\theta}}$ *as* $n \to \infty$.

This particular conjecture would hold, for example, if the l.h.s. were shown to be a continuous function of $\hat{\boldsymbol{\theta}}$ reducing monotonically and pointwise to the limit with $n$ (by Dini's theorem). We also have initial results on rate of convergence in Theorem 1 which we defer to a later publication once more fully extended.

## 5.2 Simulation of Limits and Robustness at Scale

At first sight it seems infeasible to demonstrate the above robustness and limit properties empirically on GP data-sets of size $10^6$ or above. One major obstacle being the generation of GP synthetic datasets at this size which is computationally prohibitive even allowing for the speedups described in [29]. Fortunately we can avoid the need for large-scale data-generation, in addition to achieving other major efficiencies, by adopting the approach described in Algorithm 1.

The simulation algorithm gains its efficiency by exploiting the locality of the GPnn prediction process at $\boldsymbol{x}^*$ whereby the predictive distribution only makes use of a size-$(m+1)$ marginal distribution of the full distribution of $(\boldsymbol{y}, y^*)$ over $\mathbb{R}^{n+1}$. (By the definition of a Gaussian process this marginal is a (low dimensional) multivariate Gaussian distribution from which samples can cheaply be generated). The following lemma is proved in Appendix B:

---
**Algorithm 1** Simulation of GPnn Robustness and Limiting Behaviour
---
**Input:** $n$ (training size), $n^*$ (test size), $m$ (number of nearest neighbours), $d$ ($\boldsymbol{x}$-dimension), $c(\cdot, \cdot)$ (generative covariance function), $\boldsymbol{\theta}$ (generative kernel parameters), $P(.)$ ($\boldsymbol{x}$-distribution)

**Set-up phase A:**

1. Produce the $n$ training $\boldsymbol{x}$-values and $n^*$ test $\boldsymbol{x}^*$-values by sampling independently from $P(.)$.

2. Find the $n^*$ size-$m$ nearest neighbour sets $N(\boldsymbol{x}_i^*)$ for test points $\boldsymbol{x}_i^*$ ($i = 1, \ldots, n^*$).

3. Generate $n^*$ GP samples $\boldsymbol{y}_i \in \mathbb{R}^{m+1}$ from $\mathcal{N}(0, K_{U,\boldsymbol{\theta}})$ where $U = N(\boldsymbol{x}_i^*) \cup \{\boldsymbol{x}_i^*\}$.

4. Store $(N(\boldsymbol{x}_i^*), \{y_{i,j}\}_{j=1}^m)$ which will be used for predictions in Phase B together with $y_i^* = y_{i,(m+1)} \in \mathbb{R}$, the true $y$ value at $\boldsymbol{x}_i^*$.

**Robustness Evaluation phase: B**

For several choices of assumed covariance function $\hat{c}(\cdot, \cdot)$ and parameters $\hat{\boldsymbol{\theta}}$:
For $i = 1, \ldots, n^*$ do:

1. Compute predictive distribution $\mathcal{N}(\mu_i^*, \sigma_i^{*\,2})$ at $\boldsymbol{x}_i^*$ based on $\hat{c}$, $\hat{\boldsymbol{\theta}}$ and $(N(\boldsymbol{x}_i^*), \{y_{i,j}\}_{j=1}^m)$.

2. Update NLL, MSE and calibration statistics using $\mu_i^*, \sigma_i^{*\,2}$ and $y_i^*$.

**Output:** NLL, MSE and calibration stats for range of covariance and parameter assumptions.

---

**Lemma 4** (Algorithm 1 validity). *The MSE, NLL and calibration estimates returned by Algorithm 1 are equally valid to those that would be obtained by applying the full GPnn predictive process (exactly as described in subsection 4.1) to synthetic data sets of size $n$.*

Figure 2 shows how the observed performance metrics approach the limiting behaviour as $n$ increases. In particular, from the RHS plot we see that as $n$ increases, MSE becomes increasingly insensitive to departure of $\hat{l}$ from the true value $l = 1$. This is a consequence of (what appears to be uniform) convergence of MSE toward constant value $\sigma_\xi^2 = 0.1$ (the best achievable MSE) as predicted by Theorem 1 (i). In practical terms: once a practitioner selects a particular kernel family, the accuracy of the hyperparameter $\hat{l}$ becomes less and less critical to MSE predictive accuracy with increasing $n$, so that expenditure on estimating it accurately provides diminishing returns.

The interpretation of the leftmost two plots is similar albeit somewhat more involved: The dotted red lines show the asymptotic dependence on the misspecification of the noise-variance as predicted by Theorem 1 (ii) and (iii), i.e. plots of $y = \frac{0.1}{\hat{\sigma}_\xi^2}$ and $y = \frac{1}{2}\{\log \hat{\sigma}_\xi^2 + \frac{0.1}{\hat{\sigma}_\xi^2} + \log 2\pi\}$ where $0.1 = \sigma_\xi^2$ is the true noise value used to generate the synthetic GP data. We again see evidence of uniform convergence toward this limiting behaviour with increasing $n$. The green horizontal dotted lines show the limiting values of NLL and calibration ($y = \frac{1}{2}\{\log \sigma_\xi^2 + 1 + \log 2\pi\}$ and $y = 1$ respectively) that can be achieved if the incorrect $\hat{\sigma}_\xi^2$ value is replaced by the correct value $\sigma_\xi^2$. This underlines the importance of estimating this particular parameter more accurately in order to obtain improved NLL and uncertainty calibration for large $n$. Further plots from Algorithm 1, showing dependence of each metric on all of the parameters, are given in Figure 6.

## 6 A Highly Scalable GP Nearest Neighbour Regression Algorithm

### 6.1 Parameter Estimation

**Parameter Estimation Phase 1** The first step of parameter estimation (Figure 1) involves randomly selecting a small subset $E$ of the training data to obtain a first-pass estimate $\hat{\boldsymbol{\theta}} = (\hat{l}, \hat{\sigma}_\xi^2, \hat{\sigma}_f^2)$. Small subsets yield sub-optimal $\hat{\boldsymbol{\theta}}$ values, yet as shown in 7.1, these are capable of yielding strong MSE performance due to the robustness properties of section 5. We use the method in section 3.1 of [7] to estimate parameters from $E$, randomly partitioning $E$ into $w$ size-$s$ subsets ($ws = e$) and using a block diagonal approximation (with $w$ blocks of size $s \times s$) to the full $e \times e$ gram matrix. For Table 1 we set $e = 3000, s = 300, m = 10$. For strong computational efficiency we set $e = |E|$ to a small *constant* value no matter the size of $n$. Thus as $n$ grows, an increasingly small portion of the data is used for this phase of parameter estimation and the associated cost does not increase with $n$. Note that other choices of cheap parameter estimation could be substituted here.

**Parameter Estimation Phase 2 (calibration)** As shown in section 5, NLL and calibration performance derived from $\hat{\boldsymbol{\theta}} = (\hat{l}, \hat{\sigma}_\xi^2, \hat{\sigma}_f^2)$ remain very sensitive to inaccuracies in $\hat{\sigma}_\xi^2$. An additional "calibration step" is used to refine those parameters: We randomly select a size $c$ calibration set $C$ (which is *otherwise unused*) from the training data and proceed according to Algorithm 2.

---

**Algorithm 2** Calibration of Predictive Distribution

---

**Input:** A size $c$ subset $C$ of $(\boldsymbol{x}^*, y^*)$ pairs from the training data, parameters $\hat{\boldsymbol{\theta}} = (\hat{l}, \hat{\sigma}_\xi^2, \hat{\sigma}_f^2)$.

1. For each $(\boldsymbol{x}_i^*, y_i^*) \in C$ use the efficient GPnn predictive algorithm of 6.2, with covariance function $\hat{c}(.,.)$ and parameters $\hat{\boldsymbol{\theta}} = (\hat{l}, \hat{\sigma}_\xi^2, \hat{\sigma}_f^2)$, to obtain an estimate of the mean and variance, $\mu_i^*, \sigma_i^{*\,2}$ of the predictive distribution of $y_i^*$ at $\boldsymbol{x}_i^*$.

2. Compute $\alpha = \frac{1}{c} \sum_{i=1}^c \frac{(y_i^* - \mu_i^*)^2}{\sigma_i^{*\,2}}$.

**Output:** Calibrated parameters $\hat{\boldsymbol{\theta}}' = (\hat{l}, \alpha \cdot \hat{\sigma}_\xi^2, \alpha \cdot \hat{\sigma}_f^2)$.

---

Note that this process not only adjusts the noise variance estimate $\sigma_\xi^2$ but also the kernel scale parameter $\sigma_f^2$. In so doing it simultaneously calibrates the predictive distribution and improves NLL performance whilst leaving unchanged the MSE performance obtained from the original parameter estimates $\hat{\boldsymbol{\theta}} = (\hat{l}, \hat{\sigma}_\xi^2, \hat{\sigma}_f^2)$. The lemma below is straightforward to prove (Appendix C):

**Lemma 5** (Calibration). *The parameters $\hat{\boldsymbol{\theta}}'$ output from Algorithm 2 produce GPnn predictions that (a) achieve perfect (weak) calibration on $C$, (b) minimise NLL on $C$ over all choices of $\alpha$ and (c) produce the same MSE as $\hat{\boldsymbol{\theta}}$ does on any choice of test set.*

**Remark 6.** *Algorithm 2 can be applied to other GP methods, such as SVGP, to improve calibration.*

Table 1 uses $c = 1000$; a simple refinement would be to select $c$ automatically (e.g. using a bootstrap) with optional manual override. Where accurate uncertainty calibration is paramount practitioners could devote much larger CPU resources to this phase (which is also easily distributed); when no uncertainty measures are to be used this calibration step can be bypassed altogether.

## 6.2 Efficient Nearest Neighbour Prediction

In order to implement GPnn prediction described in 4.1 we use the scikit-learn NearestNeighbors package ([22]). This implements an efficient *approximate* nearest neighbour algorithm whereby one-time work is carried out to construct a table (at $\mathcal{O}(dn \log n)$ (see e.g. [9, 21]) and counted within the total *training* times quoted) which subsequent calibration/test predictions then make use of. Query compute-costs are described in the associated documentation, e.g. $\mathcal{O}(d \log n)$ for the Ball-tree algorithm which the default automated algorithm selection in SciKit-Learn should at least match. In contrast, exact kNN costs are listed in the documentation as $\mathcal{O}(dn)$. As is evident in Table 2, Figure 3 and the quoted query and table-setup complexities, the nearest neighbour work increases with $\boldsymbol{x}$-dimension $d$. Alternative nearest neighbour algorithms and/or dimension-reduction techniques to help address this are yet to be investigated. We set the number of nearest neighbours to be $m = 400$ for all usages in this paper having observed minimal sensitivity to this parameter on independent synthetic datasets. Although a simple cross-validation procedure could be followed if tuning of $m$ is desired (at an increase in computational overhead), we wished to minimise such fine-tuning to emphasise the simplicity and robustness of the method we present, noting the strong performance we obtain despite this. At first glance, prediction complexity might appear restrictive, but some empirical tests on a laptop reveal comparable performance to SVGP prediction (Table 5).

## 7 Experimental Performance of GPnn Regression

### 7.1 Performance on Real World Datasets

**Implementational Details[1]:** Comparisons are made between our method and the state-of-the-art approaches of SVGP [14] and five distributed methods ([15, 2, 33, 7] and [18] following the recommendation in [4]). We have chosen not to include other highly-performant approximations

---

[1]https://github.com/ant-stephenson/gpnn-experiments/

Table 1: RMSE and NLL results (mean and standard deviation over 3 runs) for the best distributed method (w.r.t. MSE), SVGP and our method.

| Dataset | $n$ | $d$ | NLL Distributed | OURS | SVGP | RMSE Distributed | OURS | SVGP |
|---|---|---|---|---|---|---|---|---|
| Poletele | 4.6e+03 | 19 | $0.0091 \pm 0.015$ | **$-0.214 \pm 0.019$** | $-0.0667 \pm 0.017$ | $0.241 \pm 0.0033$ | **$0.195 \pm 0.0042$** | $0.226 \pm 0.0059$ |
| Bike | 1.4e+04 | 13 | $0.977 \pm 0.0057$ | $0.953 \pm 0.013$ | **$0.93 \pm 0.0043$** | $0.634 \pm 0.004$ | $0.624 \pm 0.0079$ | **$0.606 \pm 0.0033$** |
| Protein | 3.6e+04 | 9 | $1.11 \pm 0.0051$ | **$1.01 \pm 0.0016$** | $1.05 \pm 0.0059$ | $0.733 \pm 0.0038$ | **$0.666 \pm 0.0014$** | $0.688 \pm 0.0043$ |
| Ctslice | 4.2e+04 | 378 | $-0.159 \pm 0.052$ | **$-1.26 \pm 0.01$** | $0.467 \pm 0.016$ | $0.237 \pm 0.012$ | **$0.132 \pm 0.00062$** | $0.384 \pm 0.0064$ |
| Road3D | 3.4e+05 | 2 | $0.685 \pm 0.0041$ | **$0.371 \pm 0.004$** | $0.608 \pm 0.018$ | $0.478 \pm 0.0023$ | **$0.351 \pm 0.0014$** | $0.443 \pm 0.008$ |
| Song | 4.6e+05 | 90 | $1.32 \pm 0.0012$ | **$1.18 \pm 0.0045$** | $1.24 \pm 0.0012$ | $0.851 \pm 6.7\text{e-}05$ | **$0.787 \pm 0.0045$** | $0.834 \pm 0.0011$ |
| HouseE | 1.6e+06 | 8 | $-1.34 \pm 0.0013$ | **$-1.56 \pm 0.0065$** | $-1.46 \pm 0.0046$ | $0.0626 \pm 5.2\text{e-}05$ | **$0.0506 \pm 0.00072$** | $0.0566 \pm 0.00011$ |

(e.g. structured kernel interpolation (SKI) methods and their extensions ([37, 39, 11])), since, to our knowledge, none have supplanted these methods in the community as ubiquitous benchmarks on general datasets. Parameters for our method are given in 6.1 and 6.2. SVGP used 1024 inducing points; the distributed methods all used randomly selected subsets of sizes as close as possible to 625. The learning rate for the Adam optimiser was 0.01 for SVGP and 0.1 for our method and distributed methods. All runs in Table 1 used the the squared exponential ("RBF") covariance function. A "pre-whitening" process (E.1) was applied to x values for all methods and the y values normalised (using training data-derived means and sds) to have mean zero and variance 1. More complete details are given in E. SVGP was run on a single Tesla V100 GPU with 16GB memory; all distributed methods run on eight Intel Xeon Platinum 8000 CPUs sharing 32GB of memory. Our method was run on a Macbook Pro with 2.4 GHz Intel core i5. See D for a full explanation of our selection and pre-processing of datasets which, apart from Protein, are taken from the UCI repository.

**Results** Runs were made on three randomly selected $7/9, 2/9$ splits into training and test sets. Table 1 shows MSE and NLL results for our method alongside SVGP and distributed method (note: $n = $ *training set* size). The table shows only the best of the five distributed methods' results (w.r.t. MSE) but full results and details of all methods and all three performance measures are given in E. Complete calibration results are also plotted in Figure 4. With the exception of the Bike dataset our method is found to outperform all methods simultaneously for both MSE and NLL, and calibration likewise bar a narrow second place on the Song dataset. Table 2 and Figure 3 show that this is achieved whilst undercutting the training costs of the other methods, an effect that is very pronounced for large training sets (e.g. approximately $100\times$ faster than the other methods at $n = 1.6 \times 10^6$ on House Electric). Figure 3 shows that a significant portion of time involves calibration; this can be parallelised (or eliminated if uncertainty is not required). Note also that larger timings observed for higher dimensional datasets are due to slower performance of the approximate nearest neighbour algorithm (6.2) in that regime, both for nn table construction and calibration. As discussed in 6.2, future improvements may reduce this effect. It is very interesting that "curse-of-dimensionality" has not impacted on the method's MSE, NLL or calibration competitiveness at large $d$. This was despite the fact that a PCA analysis of the training $x$ values showed no concentration within a low dimensional space (as to be expected given the prewhitening that has been applied (subsection E.1).

**Conjecture 7.** *Robustness to "curse-of-dimensionality" is at least partially explained by the increase in the intrinsic data-length-scale by a factor of order $\sqrt{d}$ that must arise in order for GP methods to be effective.*

The heuristic reasoning behind this conjecture is as follows: Unless length scale increases with $d$ the kernel gram matrix will exhibit an abundance of exceptionally small off-diagonal entries and hence be unable able to gain significant predictive power. A $\sqrt{d}$ increase serves to counterbalance this effect and seems consistent with length scales recovered from real data in practice.

## 7.2 Performance on Massive Synthetic Datasets

We generated size $5 \times 10^7$ datasets using the 15-variable deterministic Oakley and O'Hagan function [20, 30] with i.i.d. variance-$\sigma_\xi^2$ additive noise sampled from a zero-mean Laplacian distribution (with much wider tails than $\mathcal{N}(0, \sigma_\xi^2)$). This function has 5 inputs contributing significantly to output

Table 2: Corresponding recorded training times (with mean and standard deviation from 3 runs) associated to the metrics in Table 1, i.e. recorded at the same time and with the time given for the "distributed" method relating to the best performing model in terms of MSE. Mean times are rounded to 3 s.f. and standard deviation to 2.

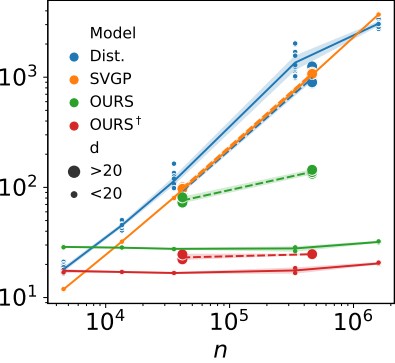

| Dataset | $n$ | $d$ | **Train time/s** Distributed | OURS | SVGP |
|---|---|---|---|---|---|
| Poletele | 4.6e+03 | 19 | 17.1 ± 0.66 | 28.8 ± 0.22 | **11.9 ± 0.081** |
| Bike | 1.4e+04 | 13 | 43.5 ± 0.64 | **28.4 ± 0.12** | 32.3 ± 0.15 |
| Protein | 3.6e+04 | 9 | 98.9 ± 1.7 | **27.7 ± 0.19** | 81.1 ± 1.1 |
| Ctslice | 4.2e+04 | 378 | 86.9 ± 1.7 | **76.1 ± 4.6** | 98.2 ± 1.8 |
| Road3D | 3.4e+05 | 2 | 1200.0 ± 110.0 | **27.9 ± 1.3** | 760.0 ± 8.0 |
| Song | 4.6e+05 | 90 | 1050.0 ± 110.0 | **138.0 ± 5.8** | 1080.0 ± 14.0 |
| HouseE | 1.6e+06 | 8 | 3110.0 ± 250.0 | **32.0 ± 0.34** | 3720.0 ± 17.0 |

Figure 3: Training times (s) for each model with "high" dimensional datasets highlighted.†: without calibration.

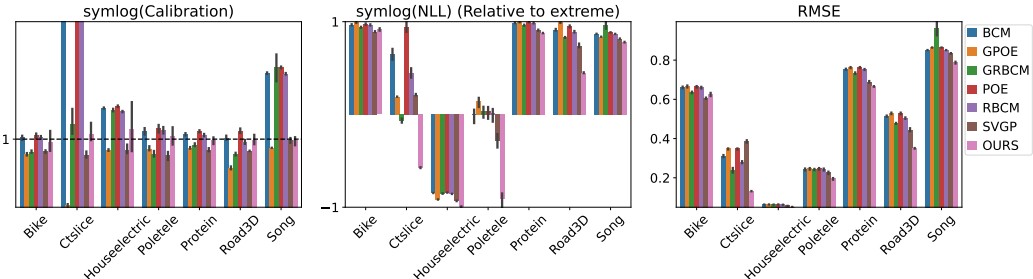

Figure 4: Experiment results on a suite of UCI datasets. Optimal calibration performance is 1 (indicated by a black dashed line). Lower is better for NLL and RMSE. Y-axis truncated for readability for Calibration due to very large values on the "Ctslice" dataset. NLL is rescaled relative to the most extreme model performance. "symlog" refers to logarithmic axis rescaling applied to the $y$-axis on both positive and negative values ("symmetric").

variance, 5 with smaller effect, and 5 with almost no effect. These properties are poorly matched by the isotropic covariance functions being applied, resulting in gross misspecification of the assumed $\mathcal{GRF}$ model and the additive noise. Figure 5 shows performance achieved with both the squared exponential ("RBF") covariance function and the exponential (Matérn 1/2) covariance function. It is very interesting to note the improvement in convergence rate achieved by the exponential kernel. (see Remark 2 for a potential explanation of why isotropic covariance functions are so effective at large $n$).

**Remark 8.** *We checked to see whether this strong exponential kernel performance extended to UCI datasets. Surprisingly, given that it is not recommended for use in GP regression (e.g. [36] page 85), it produced best RMSE performance across the board when compared with RBF and Matérn 3/2 kernels (Table 4), with Road3D RMSE reducing from 0.351 to 0.098. Exponential-kernel NLL performance was everywhere best apart from Ctslice and calibration was also better in most cases.*

## 8 Discussion

**Related work**: The basic "subset-of-data" approximation ([3]) also achieves training efficiency by using a small portion of training data and can achieve surprisingly goods results ([3], [35] section 5.1, [13]). But it typically would need a much greater proportion of training data than we are using for large $n$ due to its failure to leverage the power of large training sets for prediction; this explains why it is not consistently competitive with other methods. Passing references to the decoupling of prediction and estimation have been made, e.g. [28, 1, 3], but not shown to be as consistently powerful as we have found, nor justified in terms of robustness theory or explored as a mainstream

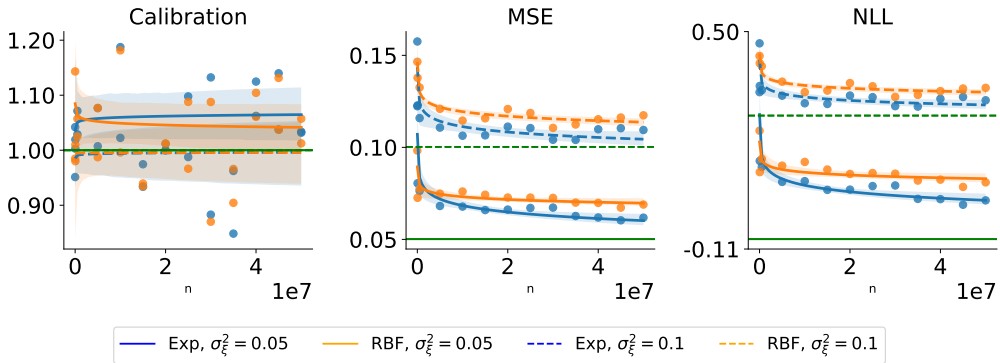

Figure 5: Behaviour of performance metrics on the Oakley and O'Hagan function-derived dataset ([20, 30]) as a function of data size, $n$. The horizontal green line indicates the limiting behaviour if the predictive and generative were to match. Shaded regions indicate 95% confidence intervals for the fitted curves.

approach. Various works (primarily from the geospatial community) make use of nearest neighbour (NN) techniques for GPs (e.g. [34, 25, 28, 6, 5, 8, 32, 38]). Vecchia ([34]) uses NNs to approximate likelihoods for parameter estimation, whilst Stein ([28]) adapts this work to REML using more distant points as well as NNs, again for parameter estimation purposes. In contrast, our focus is on using NNs for *prediction*, and whilst we have found passing references to its explicit use for this purpose (e.g. [27]) we have found little or no discussion of effectiveness in comparison to other methods on large datasets and no detailed accompanying analysis of its robustness properties and how these can achieve very high efficiency at scale. [5] gives a construction of a hierarchical fully Bayesian model ('NNGP') derived using collections of NN sets. This approach adds considerable computational overhead and code-complexity, e.g. use of Gibbs sampling. Whilst fully Bayesian treatment of hyperparameters is explored in the ML literature, e.g. [17], it has not been adopted by the ML community for use at scale due to its high computational cost relative to empirical Bayes methods ([17], section 5). Bearing this in mind, we consider the extension given in [8] for improved scalability (Algorithm 5 - 'conjugate NNGP') to be more relevant. In this hybrid method some hyperparameters are recovered as 'empirical-Bayes' point-estimates via grid-based search and the remainder treated in a fully Bayesian fashion using a conjugate prior with some choice of hyper-hyperparameters. This results in a Student-$t$ predictive distribution, rather than Gaussian, but with equal first moment to ours and variance differing only by a (hyper-hyperparameter dependent) multiplicative factor; an effect that our recalibration step would render redundant (see F.1). Recent work ([32, 38]) extends local geospatial GP methods into sparse variational ML applications. 'VNNGP' is shown to be competitive with other methods in [38] despite adding further approximations into the model. We note that when using all observations as the inducing points their predictive mechanism matches ours, up to choice of parameter estimation. We believe these pre-existing works, which differ significantly in approach and perspective, complement our own, which emphasizes the benefits of decoupling parameter estimation from prediction, the robustness properties that can be achieved at large scales, the efficacy of a simple recalibration step and can be run at high scale with a simple algorithm on an off-the-shelf laptop.

**Limitations and Future Research:** Our results exhibit a leap in speed and performance for GP regression at scale, but there remains more to be done to fully explain and extend performance (as evidenced by our remarks and conjectures). This is particularly so for high dimensional problems where (a) a faster nearest neighbour algorithm would have a particularly big pay-off and (b) there is a need to explain why "curse-of-dimensionality" appears not to have damaged the method's competitiveness (see Conjecture 7). Extensions of theory to broader aspects of GP robustness, rates of convergence and "strong calibration" (section 3) are current areas of some the authors' ongoing work.

## Acknowledgments

We would like to thank IBM Research and EPSRC for supplying iCase funding for Anthony Stephenson and the UK National Cyber Security Centre for contributing toward Robert Allison's funding. We also thank the anonymous reviewers of this paper for their constructive comments and suggestions.

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

## A Theoretical GPnn Results

### A.1 Preliminary results

Let $\rho(\boldsymbol{x}, \boldsymbol{x}') = \sigma_f \sqrt{1 - c(\boldsymbol{x}/l, \boldsymbol{x}'/l)}$ be the kernel-induced distance function over $\mathbb{R}^d$ ([23]). We define $\mathbf{x}_{(j,n)}(\boldsymbol{x}^*)$ as the $j^{th}$ nearest neighbour random variable to a test point $\boldsymbol{x}^*$ under $\rho$, which we abbreviate to $\mathbf{x}_{(j)}$ when the context is clear, and $\boldsymbol{x}_{(j)}(\boldsymbol{x}^*) \in N_m(\boldsymbol{x}^*)$ as the realised $j^{th}$ nearest neighbour of the test point $\boldsymbol{x}^*$ from a training set $X$. From this we define $\epsilon_i = \rho^2(\mathbf{x}_{(i)}, \boldsymbol{x}^*)$ and $\epsilon_{ij} = \rho^2(\mathbf{x}_{(i)}, \mathbf{x}_{(j)})$.

**Definition 9** (Support). *Let $P_{\mathbf{x}}$ be the probability measure of $\mathbf{x}$ and $S^{\rho}_{\boldsymbol{x}, \epsilon}$ the closed ball of radius $\epsilon > 0$ under the metric $\rho$ centred at $\boldsymbol{x}$. Then we define* $\mathrm{support}(P_{\mathbf{x}}) = \{\boldsymbol{x} : P_{\mathbf{x}}(S^{\rho}_{\boldsymbol{x}, \epsilon}) > 0 \, \forall \epsilon > 0\}$.

**Definition 10** (Weakly-faithful). *We define a pair of metrics $\rho(\cdot, \cdot), \hat{\rho}(\cdot, \cdot)$ to be* weakly-faithful *w.r.t. each other if the following condition holds: The $m^{th}$ nearest neighbour under $\hat{\rho}$ converges to the test point as $n \to \infty$ if and only if the $m^{th}$ nearest neighbour under $\rho$ converges to the test point in the limit.*

**Assumptions**

**(A1)** $\mathbf{x} \overset{iid}{\sim} P_{\mathbf{x}}$ and $\mathbf{x}^* \in \mathrm{support}(P_{\mathbf{x}})$ under the generative metric defined by $c(\cdot, \cdot)$.

**(A2)** $c(\cdot, \cdot), \hat{c}(\cdot, \cdot)$ are stationary kernels whose induced distance functions are *weakly faithful* metrics (Definition 10).

**(A3)** $y_i = f(\boldsymbol{x}_i) + \xi_i$ with $\xi_i \overset{iid}{\sim} P_{\xi}$, $f(\boldsymbol{x}) \sim \mathcal{WSRF}(\sigma_f^2 c(./l, ./l))$ and $y_i \mid f(\boldsymbol{x}_i) \sim P_{\xi}$ and $\mathrm{E}[\xi] = 0, \mathrm{E}[\xi^2] = \sigma_{\xi}^2$.

**Note:** Assumption (A2) is not overly restrictive and encompasses commonly used kernels such as all those mentioned in this paper.

**Lemma 11.** $\epsilon_i \to 0$ *and* $\epsilon_{ij} \to 0$ *as* $n \to \infty$ *a.e. with respect to the measure over* $\mathbf{x} \in \mathbb{R}^d$, $P_{\mathbf{x}}$, *for* $i, j \leq m$, $\frac{m}{n} \to 0$ *and under (A1-2).*

*Proof.* Lemma 6.1 of [12] states that $\|\mathbf{x}_{(m,n)}(\boldsymbol{x}) - \boldsymbol{x}\| \xrightarrow{n \to \infty} 0$ with probability one (with respect to $P_{\mathbf{x}}$). Their proof can be generalised immediately to state that $\rho(\mathbf{x}_{(m,n)}(\boldsymbol{x}), \boldsymbol{x}) \xrightarrow{n \to \infty} 0$ by using our definition of support, 9, that directly invokes the metric $\rho$. Hence $\epsilon_i \to 0$ for all $i \leq m$ (since $\boldsymbol{x}^*$ is in $\mathrm{support}(P_{\mathbf{x}})$). Since $\rho$ is a metric it satisfies the triangle inequality; hence $\rho(\mathbf{x}_{(i)}, \mathbf{x}_{(j)}) \leq \rho(\mathbf{x}_{(i)}, \boldsymbol{x}^*) + \rho(\mathbf{x}_{(j)}, \boldsymbol{x}^*) \xrightarrow{n \to \infty} 0$ for all $i, j \leq m$. $\square$

**Lemma 12.** *For an $m$-GPnn under the assumptions (A1-3),*

$$\lim_{n \to \infty} \boldsymbol{k}_N^{*T} K_N^{-1} \boldsymbol{k}_N^* = \sigma_f^2 - \sigma_{\xi}^2 m^{-1} + \mathcal{O}\left(m^{-2}\right).$$

*Proof.* From Lemma 11 we have that $\lim_{n \to \infty} k(\mathbf{x}_{(j)}(\boldsymbol{x}^*), \boldsymbol{x}^*) = \lim_{n \to \infty}(\sigma_f^2 - \epsilon_i) = \sigma_f^2$ and $\lim_{n \to \infty} k(\mathbf{x}_{(i)}(\boldsymbol{x}^*), \mathbf{x}_{(j)}(\boldsymbol{x}^*)) = \lim_{n \to \infty}(\sigma_f^2 - \epsilon_{ij}) = \sigma_f^2$. As a result, $\boldsymbol{k}_N^* \to \sigma_f^2 \mathbf{1}$ and

$$K^{\infty} := \lim_{n \to \infty} K_N = \sigma_{\xi}^2 I + \sigma_f^2 \mathbf{1} \mathbf{1}^T. \tag{8}$$

Now using Sherman-Morrison and the continuity of matrix inverse and matrix-matrix products:

$$(A + \boldsymbol{b}\boldsymbol{c}^T)^{-1} = A^{-1} - \frac{A^{-1}\boldsymbol{b}\boldsymbol{c}^T A^{-1}}{1 + \boldsymbol{c}^T A^{-1}\boldsymbol{b}} \tag{9}$$

$$(K^\infty)^{-1} = (\sigma_\xi^2 I + \sigma_f^2 \boldsymbol{1}\boldsymbol{1}^T)^{-1} = \frac{1}{\sigma_\xi^2}\left(I - \sigma_f^2 \frac{\boldsymbol{1}\boldsymbol{1}^T}{\sigma_\xi^2 + \sigma_f^2 \boldsymbol{1}^T \boldsymbol{1}}\right) \tag{10}$$

$$\boldsymbol{1}^T (K^\infty)^{-1}\boldsymbol{1} = \frac{m}{\sigma_\xi^2}\left(1 - \frac{m\sigma_f^2}{\sigma_\xi^2 + m\sigma_f^2}\right)$$

$$= \frac{m}{\sigma_\xi^2}\left(1 - m\sigma_f^2 \frac{1}{m\sigma_f^2}\left(1 - \frac{\sigma_\xi^2}{m\sigma_f^2} + \frac{\sigma_\xi^4}{m^2\sigma_f^4} - \mathcal{O}(m^{-3})\right)\right)$$

$$= \frac{1}{\sigma_f^2} - \frac{\sigma_\xi^2}{m\sigma_f^4} + \mathcal{O}(m^{-2}) . \tag{11}$$

Thus,

$$\lim_{n\to\infty} \boldsymbol{k}_N^{*T} K_N^{-1} \boldsymbol{k}_N^* = \sigma_f^4 \boldsymbol{1}^T (K^\infty)^{-1}\boldsymbol{1} = \sigma_f^2 - \sigma_\xi^2 m^{-1} + \mathcal{O}(m^{-2}) . \tag{12}$$

$\square$

**Lemma 13** ($\mathcal{WSRF}$ expectations)**.** *Under (A3),* $\mathrm{E}_{\mathbf{y},\mathrm{y}^*}\{\boldsymbol{y}y^*\} = \boldsymbol{k}^*$ *and* $\mathrm{E}_{\mathbf{y}}\{\boldsymbol{y}\boldsymbol{y}^T\} = K$.

*Proof.* By assumption on the covariance properties of $y$ and the independence and zero-mean of the additive noise, $\mathrm{E}_{\mathbf{y}}\{y_i y_j\} = k(\boldsymbol{x}_i, \boldsymbol{x}_j)$. Extending this to the joint distribution over $\boldsymbol{y}, y^*$ is straightforward and gives the results stated. $\square$

Lemma 13 is subsequently assumed to be in use throughout A.2.

## A.2 Limit proofs

In the following statements only misspecification of type (d) and/or (e) (subsection 4.2) is considered to be at work.

**Lemma 14** (MSE limit)**.** *Under the assumptions (A1-3), for fixed* $m < \infty$, *the predictive GPnn given in subsection 4.1 converges pointwise in the sense of MSE wrt* $P_{\mathbf{x}}$-*a.e. as*

$$\lim_{n\to\infty} f_n^{\mathrm{MSE}}(\boldsymbol{\theta}) = \sigma_\xi^2 (1 + m^{-1}) - \mathcal{O}(m^{-2}) .$$

*Proof.* This follows from Lemma 12 by expanding the definition of MSE:

$$\lim_{n\to\infty} f_n^{\mathrm{MSE}}(\boldsymbol{\theta}) = \lim_{n\to\infty} \mathrm{E}_{\mathbf{y},\mathrm{y}^*}\left\{|y^* - \mu_N^*|^2\right\}$$

$$= \lim_{n\to\infty}\left[\mathrm{E}_{\mathrm{y}^*}\{y^{*2}\} + \mathrm{E}_{\mathbf{y}}\{\mu_N^{*2}\} - 2\,\mathrm{E}_{\mathbf{y},\mathrm{y}^*}\{\boldsymbol{k}_N^{*T} K_N^{-1}\boldsymbol{y}_N y^*\}\right]$$

$$= \sigma_f^2 + \sigma_\xi^2 - \lim_{n\to\infty} \mathrm{E}_{\mathbf{y}}\{\mu_N^{*2}\}$$

$$= \sigma_\xi^2 (1 + m^{-1}) - \mathcal{O}(m^{-2}) .$$

Since $\mathrm{E}_{\mathbf{y}}\{\mu_N^{*2}\} = \mathrm{E}_{\mathbf{y}}\{\boldsymbol{k}_N^{*T} K_N^{-1}\boldsymbol{y}_N \boldsymbol{y}_N^T K_N^{-1}\boldsymbol{k}_N^*\} = \boldsymbol{k}_N^{*T} K_N^{-1}\boldsymbol{k}_N^*$, and by assumption $\mathrm{E}_{\mathbf{y},\mathrm{y}^*}\{\boldsymbol{y}_N y^*\} = \boldsymbol{k}_N^*$, even under a $\mathcal{WSRF}$ generative model (Lemma 13). $\square$

**Corollary 15** (NLL limit)**.**

$$\lim_{n\to\infty} f_n^{\mathrm{NLL}}(\boldsymbol{\theta}) = \frac{1}{2}\log(\sigma_\xi^2 (1 + m^{-1})) + \frac{1}{2} + \frac{1}{2}\log 2\pi - \mathcal{O}(m^{-2}) .$$

*Proof.* The proof follows straightforwardly from Lemma 12 and because $\sigma_N^{*2} = \sigma_f^2 + \sigma_\xi^2 - \boldsymbol{k}_N^{*T} K_N^{-1} \boldsymbol{k}_N^*$.

$$2\,\mathrm{E}_{\mathbf{y},\mathrm{y}^*}\left\{l_N^*\right\} = \mathrm{E}_{\mathbf{y},\mathrm{y}^*}\left\{\log \sigma_N^{*2} + \frac{(y^* - \mu_N^*)^2}{\sigma_N^{*2}} + \log 2\pi\right\}$$

$$= \log \sigma_N^{*2} + 1 + \log 2\pi$$

$$\lim_{n\to\infty} 2\,\mathrm{E}_{\mathbf{y},\mathrm{y}^*}\left\{l_N^*\right\} = \log\left(\sigma_f^2 + \sigma_\xi^2 - (\sigma_f^2 - \sigma_\xi^2 m^{-1} + \mathcal{O}(m^{-2}))\right) + 1 + \log 2\pi$$

$$= \log\left(\sigma_\xi^2(1 + m^{-1}) - \mathcal{O}(m^{-2})\right) + 1 + \log 2\pi$$

$$= \log \sigma_\xi^2 + m^{-1} + 1 + \log 2\pi - \mathcal{O}(m^{-2}).$$

$\square$

### A.2.1 Full misspecification

For the remainder of A.2 we assume that the full range of possible misspecifications ((a)-(e)) outlined in subsection 4.2 are in action. We refer to this case as "fully-misspecified" and introduce the notation $\hat{\mu}_N^*, \hat{\sigma}_N^{*2}$ to be understood to mean the predictive mean and variance under these misspecifications.

**Lemma 16** (Fully misspecified MSE limit). *For a fully misspecified model, asymptotically*

$$\lim_{n\to\infty} f_n^{\mathrm{MSE}}(\hat{\boldsymbol{\theta}}) = \sigma_\xi^2(1 + m^{-1}) \pm \mathcal{O}(m^{-2}).$$

*provided the misspecified kernel distance metric is* weakly faithful *in the sense that the $m^{th}$ nearest neighbour converges under both the true and misspecified metrics (Definition 10).*

*Proof.*

$$\mathrm{E}_{\mathbf{y}}\left\{\mathrm{E}_{\mathrm{y}^*}\left[(y^* - \hat{\mu}_N^*)^2 \mid \mathbf{y}\right]\right\} = \mathrm{E}_{\mathbf{y}}\left\{\mathrm{E}_{\mathrm{y}^*}\left[y^{*2} - 2y^*\hat{\mu}_N^* + (\hat{\mu}_N^*)^2 \mid \mathbf{y}\right]\right\}$$

$$= \mathrm{E}_{\mathbf{y}}\left\{\sigma_N^{*2} + \mu_N^{*2} - 2\mu_N^*\hat{\mu}_N^* + (\hat{\mu}_N^*)^2\right\}$$

$$= \underbrace{\sigma_N^{*2}}_{(a)} + \underbrace{\boldsymbol{k}_N^{*T} K_N^{-1} \boldsymbol{k}_N^*}_{(b)} - 2\underbrace{\boldsymbol{k}_N^{*T} \hat{K}_N^{-1} \hat{\boldsymbol{k}}_N^*}_{(c)} + \underbrace{\hat{\boldsymbol{k}}_N^{*T} \hat{K}_N^{-1} K_N \hat{K}_N^{-1} \hat{\boldsymbol{k}}_N^*}_{(d)}.$$

We can use standard results to state that $(a) + (b) = \sigma_f^2 + \sigma_\xi^2$. Then we define $\hat{\gamma} = \frac{\hat{\sigma}_f^2}{\hat{\sigma}_\xi^2 + m\hat{\sigma}_f^2}$ and expand it in terms of $m^{-1}$:

$$1 - m\hat{\gamma} = \frac{\hat{\sigma}_\xi^2}{m\hat{\sigma}_f^2} - \frac{\hat{\sigma}_\xi^4}{m^2\hat{\sigma}_f^4} + \mathcal{O}(m^{-3}).$$

In a manner similar to Lemma 12 we use this result to compute:

$$\lim_{n\to\infty}(c) = \sigma_f^2 \mathbf{1}^T \hat{\sigma}_\xi^{-2}(I - \hat{\gamma}\mathbf{1}\mathbf{1}^T)\mathbf{1}\hat{\sigma}_f^2$$

$$= \frac{\sigma_f^2\hat{\sigma}_f^2}{\hat{\sigma}_\xi^2}m(1 - m\hat{\gamma})$$

$$= \frac{\sigma_f^2\hat{\sigma}_f^2}{\hat{\sigma}_\xi^2}\left(\frac{\hat{\sigma}_\xi^2}{\hat{\sigma}_f^2} - \frac{\hat{\sigma}_\xi^4}{m\hat{\sigma}_f^4}\right) + \mathcal{O}(m^{-2})$$

$$= \sigma_f^2 - \frac{\sigma_f^2\hat{\sigma}_\xi^2}{m\hat{\sigma}_f^2} + \mathcal{O}(m^{-2})$$

and

$$\lim_{n\to\infty}(d) = \frac{\hat{\sigma}_f^4}{\hat{\sigma}_\xi^4}\mathbf{1}^T(I - \hat{\gamma}\mathbf{1}\mathbf{1}^T)(\sigma_\xi^2 I + \sigma_f^2\mathbf{1}\mathbf{1}^T)(I - \hat{\gamma}\mathbf{1}\mathbf{1}^T)\mathbf{1}$$

$$= \frac{\hat{\sigma}_f^4}{\hat{\sigma}_\xi^4}\mathbf{1}^T\left[\sigma_\xi^2 I + \sigma_f^2\mathbf{1}\mathbf{1}^T - 2\sigma_\xi^2\hat{\gamma}\mathbf{1}\mathbf{1}^T + \hat{\gamma}^2\sigma_\xi^2 m\mathbf{1}\mathbf{1}^T - 2\sigma_f^2\hat{\gamma}m\mathbf{1}\mathbf{1}^T + \sigma_f^2\hat{\gamma}^2 m^2\mathbf{1}\mathbf{1}^T\right]\mathbf{1}$$

$$= \frac{\hat{\sigma}_f^4}{\hat{\sigma}_\xi^4}m(\sigma_\xi^2 + m\sigma_f^2)\left[1 - 2m\hat{\gamma} + m^2\hat{\gamma}^2\right]$$

$$= \frac{\hat{\sigma}_f^4}{\hat{\sigma}_\xi^4}m(\sigma_\xi^2 + m\sigma_f^2)(1 - m\hat{\gamma})^2$$

$$= \frac{\hat{\sigma}_f^4}{\hat{\sigma}_\xi^4}m(\sigma_\xi^2 + m\sigma_f^2)\left(\frac{\hat{\sigma}_\xi^4}{m^2\hat{\sigma}_f^4} - 2\frac{\hat{\sigma}_\xi^6}{m^3\hat{\sigma}_f^6} + \mathcal{O}(m^{-4})\right)$$

$$= \sigma_f^2 + \frac{\sigma_\xi^2}{m} - 2\frac{\sigma_f^2}{\hat{\sigma}_f^2}\frac{\hat{\sigma}_\xi^2}{m} \pm \mathcal{O}(m^{-2}),$$

where we have used the expansion of $1 - m\hat{\gamma}$ given earlier. Putting these results together gives

$$\lim_{n\to\infty}f_n^{\mathrm{MSE}}(\hat{\boldsymbol{\theta}}) = \lim_{n\to\infty}[(a) + (b) - 2(c) + (d)]$$

$$= \sigma_f^2 + \sigma_\xi^2 - 2\left(\sigma_f^2 - \frac{\sigma_f^2\hat{\sigma}_\xi^2}{m\hat{\sigma}_f^2}\right) + \sigma_f^2 + \frac{\sigma_\xi^2}{m} - 2\frac{\sigma_f^2}{\hat{\sigma}_f^2}\frac{\hat{\sigma}_\xi^2}{m} \pm \mathcal{O}(m^{-2})$$

$$= \sigma_\xi^2(1 + m^{-1}) \pm \mathcal{O}(m^{-2}).$$

□

**Lemma 17** (Calibration limit under full misspecification).

$$\lim_{n\to\infty}f_n^{\mathrm{CAL}}(\hat{\boldsymbol{\theta}}) = \frac{\sigma_\xi^2}{\hat{\sigma}_\xi^2} \pm \mathcal{O}(m^{-2}).$$

*Proof.* We use continuity to write

$$\lim_{n\to\infty}\mathrm{E}_{\mathbf{y},y^*}\left\{\frac{(y^* - \hat{\mu}_N^*)^2}{\hat{\sigma}_N^{*2}}\right\} = \left(\lim_{n\to\infty}\frac{1}{\hat{\sigma}_N^{*2}}\right)\left(\lim_{n\to\infty}f_n^{\mathrm{MSE}}(\hat{\boldsymbol{\theta}})\right).$$

By direct application of Lemma 12 $\hat{\sigma}_N^{*2} \xrightarrow{n\to\infty} \hat{\sigma}_\xi^2(1 + m^{-1}) - \mathcal{O}(m^{-2})$ and thus

$$\lim_{n\to\infty}f_n^{\mathrm{CAL}}(\hat{\boldsymbol{\theta}}) = \frac{\sigma_\xi^2}{\hat{\sigma}_\xi^2} \pm \mathcal{O}(m^{-2}).$$

□

**Corollary 18** (NLL limit under full misspecification).

$$\lim_{n\to\infty}f_n^{\mathrm{NLL}}(\hat{\boldsymbol{\theta}}) = \frac{1}{2}\log\left(\hat{\sigma}_\xi^2(1 + m^{-1})\right) + \frac{1}{2}\frac{\sigma_\xi^2}{\hat{\sigma}_\xi^2} + \frac{1}{2}\log 2\pi \pm \mathcal{O}(m^{-2}).$$

*Proof.* We start with

$$2f_n^{\mathrm{NLL}}(\hat{\boldsymbol{\theta}}) = \mathrm{E}_{\mathbf{y},y^*}\left\{\log\hat{\sigma}_N^{*2} + \frac{(y^* - \hat{\mu}_N^*)^2}{\hat{\sigma}_N^{*2}} + \log 2\pi\right\}.$$

For the second term we use Lemma 17 so that we have

$$\lim_{n\to\infty}2f_n^{\mathrm{NLL}}(\hat{\boldsymbol{\theta}}) = \log\hat{\sigma}_\xi^2 + m^{-1} + \frac{\sigma_\xi^2}{\hat{\sigma}_\xi^2} + \log 2\pi \pm \mathcal{O}(m^{-2}).$$

□

*Proof of Theorem 1.* We construct the proof using all of the intermediate results given above. In particular item (i) follows from Lemma 16, item (ii) from Lemma 17 and item (iii) from Corollary 18.

□

# B   Validity of Algorithm 1 (Proof of Lemma 4)

*Proof of Lemma 4.* We prove the result for MSE, the proofs for NLL and calibration being essentially identical.

The proof involves showing that Algorithm 1 is exactly equivalent to Algorithm 1b, an alternative scheme which self-evidently provides valid estimates of MSE on size-$n$ synthetic datasets:

---

**Algorithm 1b** Simulation of GPnn Robustness and Limiting Behaviour (Expensive)

---

1. Generate a size-$n$ set of i.i.d. training $\boldsymbol{x}$-values $X = \{\boldsymbol{x}_i\}_{i=1}^n$ with $\boldsymbol{x}_i \sim P_{\mathbf{x}}$ ($X$ is held constant henceforth).

2. Generate $n^*$ i.i.d. test points $\boldsymbol{x}_i^* \sim P_{\mathbf{x}}$.

3. Generate $n^*$ separate independent synthetic GP samples $\boldsymbol{y}_i \in \mathbb{R}^{(n+1)}$ corresponding to each of the values $(\boldsymbol{x}_i^*, X)$.

4. From within $X$ take the $m$ nearest-neighbours $N(\boldsymbol{x}_i^*)$ to the test point $\boldsymbol{x}_i^*$.

5. Corresponding to each test point $\boldsymbol{x}_i^*$ evaluate the function $e_i^{*\prime} = f_i(\boldsymbol{y}_i) = (y_i^* - \mu_N^*(\boldsymbol{y}_i'))^2$. Where, as defined in Equation 6, $\mu_N^* = \hat{\boldsymbol{k}}_N^{*T}\hat{K}_N^{-1}\boldsymbol{y}_i'$ with $N = N(\boldsymbol{x}_i)$ and where $\boldsymbol{y}_i' \in \mathbb{R}^m$ is formed from the components of $\boldsymbol{y}_i$ corresponding to $N = N(\boldsymbol{x}_i)$.

6. Compute the average $\frac{1}{n^*}\sum_{i=1}^{n^*} e_i^{*\prime}$ to obtain the MSE statistic.

---

Algorithm 1b clearly provides a valid evaluation of the MSE arising from the full GPnn prediction process described in section 3 (albeit at prohibitive expense for large $n$). Hence it is sufficient to prove that Algorithm 1 is equivalent to Algorithm 1b. We do so by applying two minor alterations to Algorithm 1b in succession whose combined effect is to convert it to Algorithm 1. We also show that throughout this process equivalence is maintained with Algorithm 1b thus completing the proof:

Change 1: Let $\boldsymbol{y}_i = (y_i^*, \boldsymbol{y}_i', \boldsymbol{y}_i'')$ where $(y_i^*, \boldsymbol{y}_i') \in \mathbb{R}^{(m+1)}$ with $\boldsymbol{y}_i'$ corresponding to the $\boldsymbol{x}$-values in $N(\boldsymbol{x}_i^*)$. Since each function $f_i(\boldsymbol{y}_i)$ in line 5 above only depends on the components $(y_i^*, \boldsymbol{y}_i')$ of the $(n+1)$-long vector $\boldsymbol{y}_i$ we can equally write $e_i^{*\prime} = f_i(y_i^*, \boldsymbol{y}_i')$ and hence truncate each of the vectors $\boldsymbol{y}_i$ to $(y_i^*, \boldsymbol{y}_i')$ before evaluating $e_i^{*\prime}$ in 5 and this clearly leaves the output of Algorithm 1b completely unchanged.

Change 2: Note (by a standard property of multivariate normal distributions) that each of the truncated vectors $(y_i^*, \boldsymbol{y}_i')$ are i.i.d. from the multivariate normal distribution whose covariance $\Sigma$ is the kernel gram matrix corresponding to just $(\boldsymbol{x}_i^*, N(\boldsymbol{x}_i^*))$. Hence it is legitimate to change the (inefficient) way of sampling $(y_i^*, \boldsymbol{y}_i')$ (via the large sample $\boldsymbol{y}$) to direct sampling from $\mathcal{N}(\mathbf{0}, \Sigma)$ whilst still maintaining equivalence with Algorithm 1b.

In this way we arrive at Algorithm 1. $\qquad\square$

# C   Parameter Calibration (Proof of Lemma 5)

*Proof of Lemma 5.* (a) Replacing parameters $\hat{\boldsymbol{\theta}} = (\hat{l}, \hat{\sigma}_\xi^2, \hat{\sigma}_f^2)$ with $\hat{\boldsymbol{\theta}}' = (\hat{l}, \alpha\hat{\sigma}_\xi^2, \alpha\hat{\sigma}_f^2)$ changes all of the $\sigma_i^{*2}$ values to $\alpha\sigma_i^{*2}$ and therefore changes the calibration value on $C$ from $\alpha = \frac{1}{c}\sum_{i=1}^c \frac{(y_i^* - \mu_i^*)^2}{\sigma_i^{*2}}$ to $\alpha/\alpha = 1$. (b) The NLL on $C$ arising from parameters $(\hat{l}, \alpha\hat{\sigma}_\xi^2, \alpha\hat{\sigma}_f^2)$ is $\frac{1}{2c}\sum_{i=1}^c\{\log\left(\alpha\hat{\sigma}_\xi^2\right) + (y_i^* - \mu_i^*)^2/(\alpha\sigma_i^{*2}) + \log 2\pi)\}$ which, on taking first and second derivatives w.r.t. $\alpha$, is found to be uniquely minimised by $\alpha = \frac{1}{c}\sum_{i=1}^c \frac{(y_i^* - \mu_i^*)^2}{\sigma_i^{*2}}$. (c) It is easily shown that replacing parameters $(\hat{\sigma}_\xi^2, \hat{\sigma}_f^2)$ by $(k\hat{\sigma}_\xi^2, k\hat{\sigma}_f^2)$ (for any $k > 0$) in the formula for $\mu^*$ (Equation 3 and Equation 6) does not

alter $\mu^*$. Hence the value of MSE $= \frac{1}{n^*} \sum_{i=1}^{n^*} (y_i^* - \mu_i^*)^2$ on any size-$n^*$ test set is unchanged when parameters $\hat{\boldsymbol{\theta}}'$ are used in place of $\hat{\boldsymbol{\theta}}$. $\qquad\square$

# D   Real world datasets

We consider a variety of datasets from the standard UCI machine learning repository[2]. These datasets are commonly used in the GP literature (see [10] for instance) and are, in principle, easily available online. In practice, we encountered some difficulties: the dataset documentation is often limited; the dataset names commonly used in other published papers do not always match the UCI database naming and important details about data pre-processing, which features to use etc, are often omitted. There are numerous attempts on GitHub and elsewhere at cataloguing these datasets along with any pre-processing, however we had limited success using them, with many appearing unmaintained. Our focus in this work is on testing our methods on a variety of real world datasets and in a way that is, as far as possible, consistent with other papers. We therefore rejected datasets about which there is ambiguity over the correct features to use, or even which column to regress on or for which outlier rejection is required but undocumented elsewhere.

Referring to the datasets used in [10], we were able to locate the following:

- Song (`https://archive.ics.uci.edu/ml/machine-learning-databases/00203/YearPredictionMSD.txt.zip`)
- Bike (`https://archive.ics.uci.edu/ml/machine-learning-databases/00275/Bike-Sharing-Dataset.zip`)
- Poletele   (`https://archive.ics.uci.edu/ml/machine-learning-databases/parkinsons/telemonitoring/parkinsons_updrs.data`)
- Keggdirected (`https://archive.ics.uci.edu/ml/machine-learning-databases/00220/Relation%20Network%20(Directed).data`)
- Keggundirected   (`https://archive.ics.uci.edu/ml/machine-learning-databases/00221/Reaction%20Network%20(Undirected).data`)
- CTSlice   (`https://archive.ics.uci.edu/ml/machine-learning-databases/00206/slice_localization_data.zip`)
- Road3d   (`https://archive.ics.uci.edu/ml/machine-learning-databases/00246/3D_spatial_network.txt`)
- Protein   (`https://archive.ics.uci.edu/ml/machine-learning-databases/00265/CASP.csv`)
- Buzz (`https://archive.ics.uci.edu/ml/machine-learning-databases/00248/regression.tar.gz`)
- HouseElectric   (HouseE)   (`https://archive-beta.ics.uci.edu/dataset/235/individual+household+electric+power+consumption`)

We were unable to find any documentation on the Kegg datasets to indicate which of the columns should be used as the independent variable (the regressor) and neither is this mentioned in any literature of which we are aware. Initial runs of standard exact GP training and prediction produced RMSEs much higher that reported in [10]. Combining these two observations, we chose to exclude both Kegg datasets. Likewise we faced problems with Buzz. An analysis of the $y$ values revealed a small proportion of extremely large outliers that we found could unduly distort performance results (e.g. depending on whether these outliers appeared in the test set for some of the random splits). With the lack of documentation we were unable to identify an outlier rejection scheme that we were confident would be consistent with results quoted in other papers. For this reason we have excluded Buzz.

The choice of $(\boldsymbol{x}, y)$ value that we applied for each of the used datasets is as follows:

---

[2]`https://archive-beta.ics.uci.edu`, accessed April 2023.

- Song. The first column is $y$, all remaining columns are $\boldsymbol{x}$.

- Bike. We use `hour.csv`. The $y$ value is `cnt`. `dteday` (the date) is transformed to just be the integer representation of the day. `instant` is just an index so is dropped. `registered` and `casual` are dropped as `registered` + `casual` = `cnt`.

- Poletele. The $y$ value is `total_UPDRS`. The columns `subject#` and `test_time` are not relevant to the problem so are dropped.

- CTSlice. $y$ value is the final column. The first column is dropped as it is just an index. We additionally drop six columns which are constant over the majority of the dataset, namely columns 59, 69, 179, 189, 279 and 351.

- Road3d. $y$ value is the final column. The first column is dropped as it is just an index.

- Protein. This dataset was processed as per `https://github.com/hughsalimbeni/bayesian_benchmarks`, whereafter we used our own random (seeded) train/test split.

- HouseElectric. $y$ value is the column labelled "Global active power", rescaled by $1000/60$ and with "Sub metering 1,2,3" columns subtracted. We convert the date column into day-of-year/365 and the time column into time of day in minutes. Further, we remove any rows with null entries.

We note that although we are using a standard set of real-world datasets, it is not always clear exactly how others in the field have carried out their own preprocessing, limiting the ability to make direct comparisons to other results reported in the literature.

# E  Additional Implementation Details

## E.1  Pre-whitening of Data

For all datasets covered in subsection 7.1 the following "whitening" preprocessing step is adopted: Let $\boldsymbol{y}$ be the vector of all regressor values in the *training* dataset only, and $X$ the matrix of all regressands in the *training* dataset only, where each row of $X$ is a feature. Let $\mu_y, \sigma_y^2$ be the sample mean and variance of $\boldsymbol{y}$ respectively in the training dataset, then the whitened $y$ values used in both the training and test set are simply $\sigma_y^{-1}(y - \mu_y)$. Let $\boldsymbol{\mu}_X, \Sigma_X$ be the sample mean and covariance matrix of $X$ respectively . Let $\Sigma_X = MM^T$, then the whitened $x$ values in both the training and test data are $\frac{1}{\sqrt{d}}M^{-1}(\boldsymbol{x} - \boldsymbol{\mu}_X)$, where $d$ is the feature dimension of $X$. **Note**: the performance metrics given in subsection 7.1 are expressed in terms of the whitened $y$ values rather than the $y$ values in their original form. This appears to be common practice in the literature and has no bearing on the *comparative* performance of the different methods within this paper.

## E.2  Test-Set Batching

To prevent excessive memory consumption, we perform all predictions for the distributed and variational methods in batches of 1000 points at a time. Where this is not possible (e.g. for especially large datasets), we use smaller batches of 500 or 250 points, as appropriate.

## E.3  Additional Implementation Details for SVGP

We use the sparse variational inducing point approach of [14], following the implementation provided by GPyTorch, which in particular uses a Choleksy decomposition to parameterise the covariance matrix of the variational prior. We broadly follow the SVGP implementation example provided by `https://docs.gpytorch.ai/en/stable/examples/04_Variational_and_Approximate_GPs/SVGP_Regression_CUDA.html`. In particular, we follow their example in using the Adam optimiser to train our model over 100 epochs with a minibatch size of 1024 and a learning rate of 0.01. We opt to use 1024 inducing points. All experiments under this method are run on a SageMaker ml.p3.2xlarge instance, consisting of a single Tesla V100 GPU with 16GB of memory.

### E.4 Additional Implementation Details for Distributed methods

A good introduction to distributed methods for Gaussian process inference is [7]. Here we run the product-of-experts (PoE) [15], generalised product-of-experts (gPoE) [2], Bayesian committee machine (BCM) [33], robust Bayesian committee machine (rBCM) [7] and generalised robust Bayesian committee machine (GrBCM) [18] following the recommendation in [4] to aggregate in $f$-space. There are three components to any distributed method: the hyperparameter inference, the *partitioner* and the *aggregator*. Hyperparameter estimation is the same for all of the methods: we use the method in section 3.1 of [7], randomly partitioning the entire training set into subsets of size 625 (or as close as possible with equal-sized experts given that in general $n$ is not a multiple of 625). A block diagonal approximation (with $n/625$ blocks) is then used to approximate to the full $n \times n$ gram kernel matrix. To recover hyperparameters with this we use Gaussian Process models with a zero prior mean and a scaled square-exponential kernel. Training is conducted using the Adam optimiser with a learning rate of 0.1 over 100 optimiser iterations. Once the hyperparameters are trained, we run our distributed prediction mechanism to evaluate performance against the test-set. The 625-sized partitioned blocks are referred to as "experts" and the shared hyperparameter values are distributed to each expert and held fixed thereafter. In the *aggregator*, or distributed prediction phase, each expert produces an individual predictive distribution and these are then aggregated to a final predictive mean and variance for each of our test points. GRBCM prediction is a little more complex than this as it makes use of an additional "communications" expert as explained in [18], aggregating in $f$-space as recommended in [4]. We provide timing statistics for training these models.

We use our own GPyTorch-based implementation of distributed GP approximations. All exact GP calculations are performed using GPyTorch using the default settings (so 20 Lanczos iterations throughout and a CG tolerance of 1 for hyperparameter inference, and $10^{-3}$ for posterior predictions). For all of our experiments, we utilise an AWS t3.2xlarge instance (consisting of 8 Intel Skylake Processors and 32 GB of RAM).

### E.5 Reproducibility

All code used to generate tables and figures in this document can be found at `https://github.com/ant-stephenson/gpnn-experiments/`.

## F  Related Work

### F.1  NNGP

An hierarchical Bayesian approach to nearest neighbour GPs is derived in [5], who construct a full stochastic process allowing an end-to-end probabilistic approach they term 'NNGP' derived from a 'parent' GP using collections of nearest neighbour sets forming a 'reference' set. This work is modified in various ways in [8] including by adaption to a hybrid empirical-Bayes/fully-Bayesian approach to improve scalability. This is described in Algorithm 5 which presents an MCMC-free approach ('conjugate NNGP') where some parameters are estimated via K-fold cross-validation and the remainder are given conjugate priors to allow exact posterior inference. Under this model, the marginal predictive distribution is Student-$t$ with mean and variance expressions that match that given by GPnn up to a scaling factor on the variance. Here we will go into more detail to make this connection explicit. First of all, note that they use an alternative parameterisation to us, specifying an inverse lengthscale ($\phi$), a ratio ($\alpha$) of noise variance ($\tau^2$) to kernelscale ($\sigma^2$) and a vector of coefficients for their linear mean function ($\boldsymbol{\beta}$, which we can neglect since we focus exclusively on mean-zero GPs).

In Algorithm 5 they go on to give the following expressions for the predictive mean and variance:

$$z = M(s, N(s, k))$$
$$w = \text{solve}(M[N(s, k), N(s, k)], z)$$
$$m_0 = \widehat{y(s)} = \text{dot}(x(s), g(k)) + \text{dot}(w, (y[N(s, k)] - \text{dot}(X[N(s, k), ], g(k))))$$
$$u = x(s) - \text{dot}(X[N(s, k), ], w)$$
$$v_0 = \text{dot}(u, \text{gemv}(V(k), u)) + 1 + \alpha - \text{dot}(w, z)$$
$$\widehat{\text{Var}(y(s))} = b_\sigma^*(k) v_0 / (a_\sigma^*(k) - 1)$$

We can see that the predictive mean $m_0$ matches our own in the mean-zero setting. In particular, in this case the only non-zero term is $\text{dot}(w, y[N(s, k)])$ which can be re-written in our notation as $\boldsymbol{k}_N^{*T} K_N^{-1} \boldsymbol{y}_N$. Note that this expression is, for fixed $\alpha$, independent of choice of kernelscale. This can be seen using the notation given in section 1: $\boldsymbol{k}_N^{*T} K_N^{-1} \boldsymbol{y}_N = \sigma_f^2 \boldsymbol{c}_N^{*T} (\sigma_f^2 C_N + \sigma_\xi^2 I)^{-1} \boldsymbol{y}_N = \boldsymbol{c}_N^{*T} (C_N + \alpha I)^{-1} \boldsymbol{y}_N$.

In addition to this, the predictive variance is equivalent to ours up to multiplicative scaling $S = b_\sigma^*(k)/[\sigma_f^2(a_\sigma^*(k) - 1)]$; i.e. $\widehat{\text{Var}(y(s))} = S\sigma_N^{*2}$, which be seen as follows: In the mean-zero case we have $v_0 = 1 + \alpha - \text{dot}(w, z)$ which can again be re-written in our notation as $1 + \sigma_\xi^2/\sigma_f^2 - \boldsymbol{c}_N^{*T} C_{\alpha,N}^{-1} \boldsymbol{c}_N^*$ (where $C_{\alpha,N} = C_N + \alpha I$). Hence, if we rescale this by a factor of $\sigma_f^2$ we exactly obtain the GPnn predictive variance, so that $\widehat{\text{Var}(y(s))} = b_\sigma^*(k) v_0 / (a_\sigma^*(k) - 1) = b_\sigma^*(k) \sigma_N^{*2} / [\sigma_f^2(a_\sigma^*(k) - 1)]$ as claimed. Since GPnn uses an additional recalibration step to rescale all predictive variances by a single shared multiplicative factor, the factor $S$ between the two methods can be made effectively redundant: i.e. if our recalibration step (Algorithm 2) were applied to their method, the two methods would become equivalent *with respect to RMSE and weak-calibration* (setting aside differences in parameter estimation and their associated costs).

We reemphasise that whilst GPnn and Conjugate NNGP pointwise predictions can be related as above, the latter is given in the form of a $t$-distribution which, even with matched first two moments, will have a different shape to that of a Normal distribution. As such, the (Gaussian) NLL performance measure is no longer a valid choice to assess this model. RMSE and weak-calibration remain distribution-agnostic however.

# G   Further simulation results

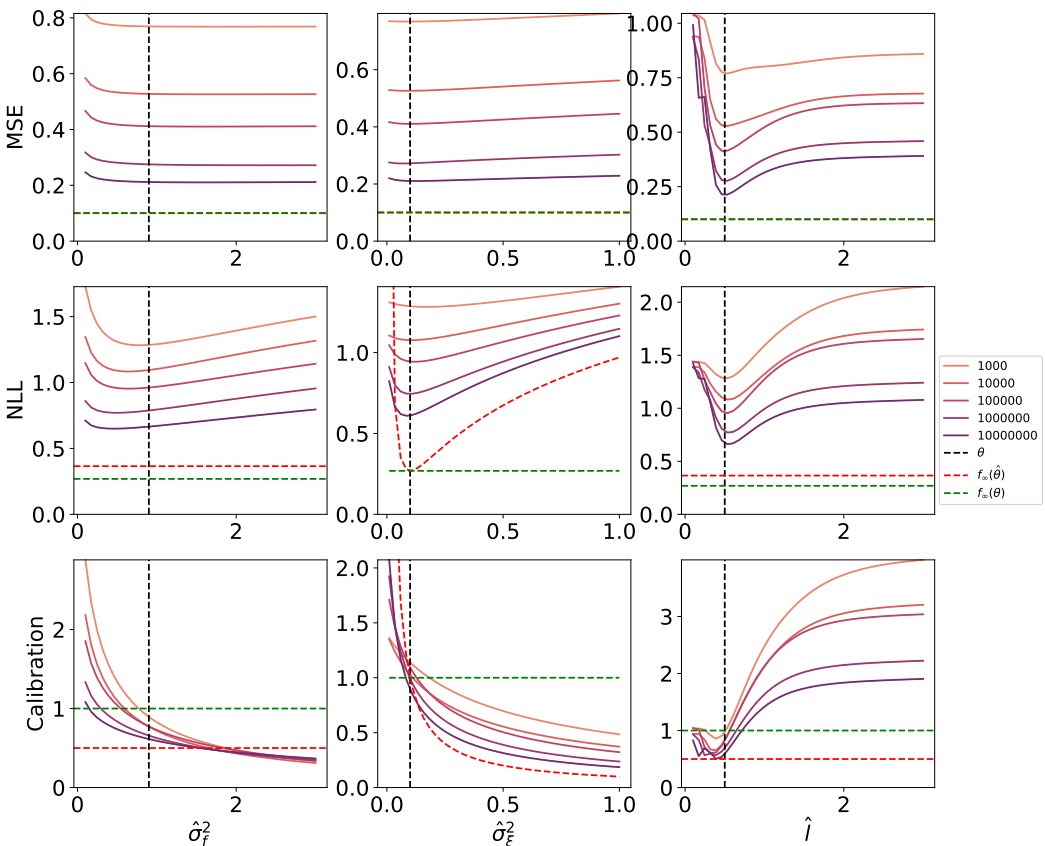

Figure 6: Behaviour of performance metrics as functions of kernel hyperparameters for increasing training set sizes $n$. The black dashed line denotes the true parameter value; the red dashed line shows the limiting behaviour as $n \to \infty$ and the green dashed line shows the limiting behaviour when the hyperparameters are correct. Simulations run with $d = 20, l = 0.5, \sigma_\xi^2 = 0.1, \sigma_f^2 = 0.9$. Assumed parameters when constant: $\hat{\sigma}_\xi^2 = 0.2, \hat{\sigma}_f^2 = 0.8, \hat{l} = 0.5$.

# H  Further Results on UCI Datasets

## H.1  Results for all distributed methods

Table 3: Results for all methods on all metrics.

| Dataset | $n$ | $d$ | Model | Calibration | NLL | RMSE |
|---|---|---|---|---|---|---|
| Bike | 1.4e+04 | 13 | BCM | 1.02 ± 0.02 | 1.0 ± 0.0065 | 0.66 ± 0.0043 |
| | | | GPOE | 0.873 ± 0.012 | 1.03 ± 0.0069 | 0.664 ± 0.0054 |
| | | | GRBCM | 0.893 ± 0.014 | 0.977 ± 0.0057 | 0.634 ± 0.004 |
| | | | OURS | 0.974 ± 0.087 | 0.953 ± 0.013 | 0.624 ± 0.0079 |
| | | | POE | 1.03 ± 0.022 | 1.01 ± 0.0083 | 0.664 ± 0.0054 |
| | | | RBCM | **1.01 ± 0.02** | 1.0 ± 0.0065 | 0.659 ± 0.0043 |
| | | | SVGP | 0.898 ± 0.011 | **0.93 ± 0.0043** | **0.606 ± 0.0033** |
| Ctslice | 4.2e+04 | 378 | BCM | 5.04 ± 0.28 | 1.43 ± 0.13 | 0.311 ± 0.0052 |
| | | | GPOE | 0.435 ± 0.013 | 0.422 ± 0.0015 | 0.347 ± 0.0027 |
| | | | GRBCM | 1.13 ± 0.11 | -0.159 ± 0.052 | 0.237 ± 0.012 |
| | | | OURS | **1.04 ± 0.085** | **-1.26 ± 0.01** | **0.132 ± 0.00062** |
| | | | POE | 6.39 ± 0.27 | 2.08 ± 0.12 | 0.347 ± 0.0027 |
| | | | RBCM | 4.16 ± 0.25 | 0.987 ± 0.11 | 0.28 ± 0.0048 |
| | | | SVGP | 0.865 ± 0.026 | 0.467 ± 0.016 | 0.384 ± 0.0064 |
| Houseelectric | 1.6e+06 | 8 | BCM | 1.27 ± 0.0046 | -1.33 ± 0.0009 | 0.0634 ± 3.5e-05 |
| | | | GPOE | 0.908 ± 0.0065 | -1.43 ± 0.0016 | 0.0638 ± 7.7e-05 |
| | | | GRBCM | 1.25 ± 0.011 | -1.34 ± 0.0039 | 0.063 ± 0.00026 |
| | | | OURS | **1.08 ± 0.21** | **-1.56 ± 0.0065** | **0.0506 ± 0.00072** |
| | | | POE | 1.28 ± 0.006 | -1.32 ± 0.0018 | 0.0638 ± 7.7e-05 |
| | | | RBCM | 1.24 ± 0.0054 | -1.34 ± 0.0013 | 0.0626 ± 5.2e-05 |
| | | | SVGP | 0.911 ± 0.038 | -1.46 ± 0.0046 | 0.0566 ± 0.00011 |
| Poletele | 4.6e+03 | 19 | BCM | 1.07 ± 0.029 | 0.00035 ± 0.019 | 0.243 ± 0.0048 |
| | | | GPOE | 0.917 ± 0.02 | 0.0344 ± 0.013 | 0.246 ± 0.0038 |
| | | | GRBCM | 0.872 ± 0.024 | 0.0091 ± 0.015 | 0.241 ± 0.0033 |
| | | | OURS | **1.03 ± 0.073** | **-0.214 ± 0.019** | **0.195 ± 0.0042** |
| | | | POE | 1.1 ± 0.036 | 0.00772 ± 0.016 | 0.246 ± 0.0038 |
| | | | RBCM | 1.08 ± 0.029 | 0.00309 ± 0.018 | 0.243 ± 0.0048 |
| | | | SVGP | 0.862 ± 0.035 | -0.0667 ± 0.017 | 0.226 ± 0.0059 |
| Protein | 3.6e+04 | 9 | BCM | 1.04 ± 0.0097 | 1.14 ± 0.003 | 0.754 ± 0.0022 |
| | | | GPOE | 0.925 ± 0.007 | 1.15 ± 0.0035 | 0.763 ± 0.0024 |
| | | | GRBCM | 0.95 ± 0.012 | 1.11 ± 0.0051 | 0.733 ± 0.0038 |
| | | | OURS | **0.991 ± 0.029** | **1.01 ± 0.0016** | **0.666 ± 0.0014** |
| | | | POE | 1.07 ± 0.0088 | 1.15 ± 0.0033 | 0.763 ± 0.0024 |
| | | | RBCM | 1.03 ± 0.0096 | 1.13 ± 0.003 | 0.752 ± 0.0022 |
| | | | SVGP | 0.908 ± 0.016 | 1.05 ± 0.0059 | 0.688 ± 0.0043 |
| Road3D | 3.4e+05 | 2 | BCM | 1.01 ± 0.017 | 0.753 ± 0.007 | 0.514 ± 0.0035 |
| | | | GPOE | 0.756 ± 0.012 | 0.819 ± 0.0054 | 0.529 ± 0.0037 |
| | | | GRBCM | 0.873 ± 0.011 | 0.685 ± 0.0041 | 0.478 ± 0.0023 |
| | | | OURS | **0.991 ± 0.041** | **0.371 ± 0.004** | **0.351 ± 0.0014** |
| | | | POE | 1.07 ± 0.019 | 0.783 ± 0.0076 | 0.529 ± 0.0037 |
| | | | RBCM | 0.976 ± 0.016 | 0.735 ± 0.0066 | 0.505 ± 0.0034 |
| | | | SVGP | 0.9 ± 0.00094 | 0.608 ± 0.018 | 0.443 ± 0.008 |
| Song | 4.6e+05 | 90 | BCM | 1.56 ± 0.0063 | 1.32 ± 0.0012 | 0.851 ± 6.7e-05 |
| | | | GPOE | 0.926 ± 0.00049 | 1.27 ± 3.4e-05 | 0.864 ± 7.5e-05 |
| | | | GRBCM | 1.61 ± 0.11 | 1.46 ± 0.058 | 0.961 ± 0.035 |
| | | | OURS | 0.99 ± 0.037 | **1.18 ± 0.0045** | **0.787 ± 0.0045** |
| | | | POE | 1.61 ± 0.0067 | 1.34 ± 0.0013 | 0.864 ± 7.5e-05 |
| | | | RBCM | 1.56 ± 0.0062 | 1.31 ± 0.0011 | 0.851 ± 6.4e-05 |
| | | | SVGP | **0.991 ± 0.02** | 1.24 ± 0.0012 | 0.834 ± 0.0011 |

## H.2 Performance of different kernels

Table 4: Results on the UCI datasets using different kernel choices for our method and demonstrating the apparent superiority of the exponential kernel in these cases.

| Dataset | $n$ | $d$ | Calibration Distributed | Ours (Exp) | Ours (Matérn) | Ours (RBF) | SVGP |
|---|---|---|---|---|---|---|---|
| Poletele | 4.6e+03 | 19 | 0.872 ± 0.024 | **0.994 ± 0.15** | 0.971 ± 0.13 | 1.03 ± 0.073 | 0.862 ± 0.035 |
| Bike | 1.4e+04 | 13 | 0.893 ± 0.014 | **0.988 ± 0.098** | 0.971 ± 0.086 | 0.974 ± 0.087 | 0.898 ± 0.011 |
| Protein | 3.6e+04 | 9 | 0.95 ± 0.012 | **0.995 ± 0.038** | 0.993 ± 0.031 | 0.991 ± 0.029 | 0.908 ± 0.016 |
| Ctslice | 4.2e+04 | 378 | 1.13 ± 0.11 | 0.912 ± 0.071 | **1.04 ± 0.082** | 1.04 ± 0.085 | 0.865 ± 0.026 |
| Road3D | 3.4e+05 | 2 | 0.873 ± 0.011 | 1.09 ± 0.065 | **1.0 ± 0.054** | 0.991 ± 0.041 | 0.9 ± 0.00094 |
| Song | 4.6e+05 | 90 | 1.56 ± 0.0063 | **0.995 ± 0.033** | 0.994 ± 0.035 | 0.99 ± 0.037 | 0.991 ± 0.02 |
| Houseelectric | 1.6e+06 | 8 | 1.24 ± 0.0054 | 1.11 ± 0.29 | **1.08 ± 0.27** | 1.08 ± 0.21 | 0.911 ± 0.038 |

| Dataset | $n$ | $d$ | RMSE Distributed | Ours (Exp) | Ours (Matérn) | Ours (RBF) | SVGP |
|---|---|---|---|---|---|---|---|
| Poletele | 4.6e+03 | 19 | 0.241 ± 0.0033 | **0.169 ± 0.0076** | 0.17 ± 0.0076 | 0.195 ± 0.0042 | 0.226 ± 0.0059 |
| Bike | 1.4e+04 | 13 | 0.634 ± 0.004 | **0.565 ± 0.0036** | 0.6 ± 0.0044 | 0.624 ± 0.0079 | 0.606 ± 0.0033 |
| Protein | 3.6e+04 | 9 | 0.733 ± 0.0038 | **0.58 ± 0.0068** | 0.629 ± 0.004 | 0.666 ± 0.0014 | 0.688 ± 0.0043 |
| Ctslice | 4.2e+04 | 378 | 0.237 ± 0.012 | **0.123 ± 0.004** | 0.126 ± 0.0024 | 0.132 ± 0.00062 | 0.384 ± 0.0064 |
| Road3D | 3.4e+05 | 2 | 0.478 ± 0.0023 | **0.0976 ± 0.013** | 0.27 ± 0.01 | 0.351 ± 0.0014 | 0.443 ± 0.008 |
| Song | 4.6e+05 | 90 | 0.851 ± 6.7e-05 | **0.776 ± 0.004** | 0.778 ± 0.0045 | 0.787 ± 0.0045 | 0.834 ± 0.0011 |
| Houseelectric | 1.6e+06 | 8 | 0.0626 ± 5.2e-05 | **0.045 ± 0.00025** | 0.0485 ± 0.0004 | 0.0506 ± 0.00072 | 0.0566 ± 0.0001 |

| Dataset | $n$ | $d$ | NLL Distributed | Ours (Exp) | Ours (Matérn) | Ours (RBF) | SVGP |
|---|---|---|---|---|---|---|---|
| Poletele | 4.6e+03 | 19 | 0.0091 ± 0.015 | **-0.397 ± 0.028** | -0.346 ± 0.032 | -0.214 ± 0.019 | -0.0667 ± 0.017 |
| Bike | 1.4e+04 | 13 | 0.977 ± 0.0057 | **0.854 ± 0.004** | 0.915 ± 0.0077 | 0.953 ± 0.013 | 0.93 ± 0.0043 |
| Protein | 3.6e+04 | 9 | 1.11 ± 0.0051 | **0.853 ± 0.013** | 0.95 ± 0.0061 | 1.01 ± 0.0016 | 1.05 ± 0.0059 |
| Ctslice | 4.2e+04 | 378 | -0.159 ± 0.052 | -1.05 ± 0.027 | **-1.31 ± 0.017** | -1.26 ± 0.01 | 0.467 ± 0.016 |
| Road3D | 3.4e+05 | 2 | 0.685 ± 0.0041 | **-0.931 ± 0.14** | 0.109 ± 0.039 | 0.371 ± 0.004 | 0.608 ± 0.018 |
| Song | 4.6e+05 | 90 | 1.32 ± 0.0012 | **1.16 ± 0.0046** | 1.17 ± 0.0051 | 1.18 ± 0.0045 | 1.24 ± 0.0012 |
| Houseelectric | 1.6e+06 | 8 | -1.34 ± 0.0013 | **-1.95 ± 0.028** | -1.62 ± 0.0095 | -1.56 ± 0.0065 | -1.46 ± 0.0046 |

## H.3 Prediction times

Table 5: Prediction times (in seconds) for GPnn and SVGP with 400 nearest-neighbours and 1024 inducing points respectively, over a small range of dataset sizes and dimensions.

| $n$ | $d$ | GPnn | SVGP |
|---|---|---|---|
| 4.2e4 | 378 | 0.06 | 0.02 |
| 3.6e4 | 9 | 0.02 | 0.02 |
| 1.1e5 | 50 | 0.03 | 0.06 |
| 1.1e5 | 10 | 0.02 | 0.06 |

# I  Overall Computational Expenditure

Our distributed and variational method experiments were conducted using cloud computing resources. Experiments using our own method have been carried out on an author's laptop. SVGP experiments were run using a SageMaker virtual machine on a single Nvidia Tesla V100 GPU with 16GB memory. Distributed method experiments were run using eight Intel Xeon Platinum 8000 CPU cores (t3.2xlarge EC2 instances).

Below we will attempt to give reasonable indications of the amount of computational work expended to obtain the results in this paper, though note that we are neglecting the work expended in the development and research stages that did not directly contribute to the runs in the paper. As such, the costs presented are representative of the costs of replicating our paper, not repeating the research from scratch. Instead of reporting costs in dollars, we will report approximate computing hours for each instance type. The reader can then estimate their own costs using the current instance costs in the region of their choice, or under other cloud providers or even using on-premise compute.

| Dataset | Billed hours (1 GPU, 3 runs) | Billed hours (8 CPUs, 3 runs of 5 methods) |
|---|---|---|
| bike | 0.027 | 0.222 |
| ctslice | 0.082 | 0.546 |
| houseelectric | 3.713 | 256.776 |
| poletele | 0.010 | 0.079 |
| protein | 0.068 | 0.680 |
| road3d | 0.635 | 31.674 |
| song | 0.904 | 12.237 |

This gives a total of around 5.4 hours of compute time on a 1 GPU VM and 302.2 hours on an 8 CPU VM.

