# OpenReview forum: "Leveraging Locality and Robustness to Achieve Massively Scalable Gaussian Process Regression"
_NeurIPS.cc/2023/Conference — NeurIPS 2023 poster_

### Official Review · Reviewer_3iYQ · 2023-07-05

**Soundness:** 3 good
**Presentation:** 3 good
**Contribution:** 3 good
**Rating:** 7
**Confidence:** 3

**Summary:**

The paper proposes a Gaussian process regression technique that only uses nearest neighbors for prediction. The question of choosing hyperparameters is addressed and asymptotic behavior is analyzed. The method shows substantial speed up on UCI datasets while also providing improved predictive performance.

**Strengths:**

The empirical evaluation shows very good results. Even so, the method is theoretically analyzed and discussed with interesting insights and hypotheses. The paper is well-written.

**Weaknesses:**

A comparison with other nearest neighbor based techniques might have been helpful (e.g. ones cited in the paper). Also, it might make sense to investigate more into the geostatistical work on the subject and cite something from the area. I know that nearest neighbor based predictions are quite often considered to be standard in the area.

**Questions:**

Due to time constraints and NeurIPS not accommodating my request for a reduced reviewing workload, I did not have the opportunity to thoroughly review the proofs or examine all the details.

**Limitations:**

The limitations are addressed.

---

> ### Author Rebuttal · Authors · 2023-08-08
>
> 1. Thank you for what time you had available and your comments. We take your point regarding references to other work on similar topics and intend to expand our section on "related work" to reflect this, including contributions from the geospatial community.
> 2. We acknowledge that a comparison to other nearest-neighbour based methods might have been interesting, but we neglected to do so on the basis that the thrust of this paper was to emphasise, for example, the following points:
> - The decoupling of parameter estimation and prediction and the associated advantages to computational cost this can incur
> - The asymptotic insensitivity of GPnn prediction to model misspecification making use of nearest-neighbours to do so, rather than the focus being primarily on NNs itself.
> 3. Please also see our response to reviewer MqBs bullet point 4 concerning our choice of "state of the art" comparisons.

---

> > ### Comment · Reviewer_3iYQ · 2023-08-14
> >
> > I have read the rebuttal. My score remains unchanged.
> >
> > I suggest the authors to compare at least against https://www.ncbi.nlm.nih.gov/pmc/articles/PMC5927603/, at least describing the differences between the mentioned paper and their paper in a sentence or two: the cited work is quite well-known and thus I think such a comparison would benefit the readers.

---

> > > ### Author Response · Authors · 2023-08-15
> > >
> > > Many thanks for this response. We agree that this would be a good paper to reference and briefly compare and contrast with our approach. We will implement your suggestion.

---

### Official Review · Reviewer_N8D3 · 2023-07-06

**Soundness:** 2 fair
**Presentation:** 2 fair
**Contribution:** 2 fair
**Rating:** 3
**Confidence:** 3

**Summary:**

The paper proposes a change in perspective in how Gaussian process regression can be leveraged by conditioning predictive distribution on only the neighboring data points. The paper argues against the common practice of using one singular set of data points for model hyperparameter estimation and prediction, and presents a framework for running scalable approximation of GP regression using locality information.

**Strengths:**

I believe that the overall idea of leveraging locality in Gaussian process models has a lot of potential: while I am skeptical as the arguments in the paper seem unconvincing, I think there is merit in good empirical performance for such algorithms. However, I believe that a comparison with existing methods should be done in a more proper manner.

**Weaknesses:**

While using KNN in combination of GP regression might seem like a straightforward idea to massively speed up the inference of GP models, I believe that many key aspects of GPs are overlooked, constituting a major weakness from this paper. I list some of the specific examples below.
- Combining KNN with GP is overall not a novel idea, the practice is not exactly new. A cursory search reveals that many research papers outside the machine learning community are already using it as a proxy for the less scalable fullscale GP regression. Some examples relevant to machine learning include: 1. Wu L, Pleiss G, Cunningham JP. Variational nearest neighbor Gaussian process. In: Proceedings of the 39th International Conference on Machine Learning, PMLR; 2022; 2. Chen H, Zheng L, Kontar RA, Raskutti G. Gaussian Process Parameter Estimation Using Mini-batch Stochastic Gradient Descent: Convergence Guarantees and Empirical Benefits. Journal of Machine Learning Research. 2022;23(227):1–59.
- The paper focuses on _pointwise_ uncertainty prediction, while GP posteriors provide not only a pointwise uncertainty, but _pairwise_ covariance as well. The covariance kernel is such a crucial element that defines the hypothesis space of GP models that given a configuration of pointwise uncertainty, there exists countless GPs that takes identical pointwise variance.
- While KNN is guaranteed to speed up massively in lower dimensions, it suffers from curse of dimensionality itself as it becomes harder to find nearest neighbors.
- While it is not mandatory for parameter estimation and prediction to be conditioned on the same set of data, SVGP does not incur much computational expense at prediction level. Conventional GP regression and SVGP only need one matrix inversion operation to predict the uncertainty level on an arbitrary number of test points, while GPnn needs a matrix inversion operation for every separate data point -- essentially it becomes a balancing act of whether optimizing an SVGP or apply KNN to every test point is more expensive.
- Theorem 1 discusses the asymptotic behavior as $n\rightarrow\infty$. While the result of insensitivity w.r.t. kernel parameter carries some value, the _rate_ of convergence matters in a finite data regime. For example, Ackermann function and inverse Ackermann function both converge to infinity but at drastically different rates.
- While the common squared exponential kernel is designed to look at neighborhood information, many more expressive kernels can have longer-range connections. I think the paper tacitly admits that GPnn can only be used for interpolation tasks, but it is somewhat unfair to state that kernel hyperparameter does not matter as it is always a misspecified model.
- The ease at obtaining easy uncertainty information given neighborhood information is conditioned on conjugate likelihood, with a Gaussian observational model. As some degree of variational inference is required for non-conjugate likelihood, GPnn struggles to apply beyond regression tasks.
- Simulating synthetic data using Algorithm 11 seems flawed. Algorithm 1 generates test points as if every 2 test points are un-correlated, which naturally gives GPnn an advantage.

**Questions:**

None.

**Limitations:**

I have listed the limitations in the "weakness" section.

---

> ### Author Rebuttal · Authors · 2023-08-08
>
> 1. We briefly mention use of NN in other GP literature in sec. 8 "related work" but appreciate your feedback pointing to the need to expand that further, including how our use differs from theirs. We agree that  the mere act of including kNN in some form within a paper cannot be regarded as novel. However, the novelty is in how we use it, the resulting strength of results and the substantial computational efficiency savings made. Novel aspects include:  robustness in the limit to kernel choice and hyperparameters, substantial reductions in training costs, justification of detachment of prediction process from training process, improvements in calibration over other methods, algorithmic simplicity. These innovations are more general than use of NN alone.
>
> 2. Thank you for this perspective. We have focused on pointwise prediction and, in light of your feedback, will clarify that in the paper. Whilst accepting pointwise prediction does not cover the entire spectrum of GP applications it is an exceptionally important area in its own right (and the most heavily used in practice) for which nearly all research papers quote their performance results, e.g. [2,4,7,11,12,13,22,24,25]. The importance of making improvements in this area is also reflected in the comments of all reviewers of this paper. Selling points of GPs in the pointwise context are provision of accuracy and well principled uncertainty measures. Our method is shown to generally outperform the other methods in both respects at a small fraction of their training cost.
>
> 3. Prior to our analysis, we too had thought that curse of dimensionality would impact badly on comparative performance for high-dimensional datasets. Surprisingly, we found mse, nll and calibration still beat other methods at large d (e.g. Ctslice with d=378). This led to conjecture 6 and related follow-on work in progress. Computational cost does rise with d, as covered in 6.2, table 2 and figure 3, but even for the most extreme case (d=378) GPnn outperforms other methods on training time and is comparable at test time.  Given that the precise implementation of the approximate kNN algorithm is beyond the scope of this paper, and should not negatively affect the locality arguments on which this work is structured, we believe that advances in this area will only improve the performance beyond its already strong baseline.
>
> 4. Whilst true that SVGP prediction is fast, GPnn prediction is comparable (see table 1 in the pdf) and training time much faster than that of SVGP, meaning for a given computational budget covering both train and test time GPnn is very competitive, especially at large n. On test point timings, table 1 gives exemplar prediction times in seconds, all obtained on a laptop (with 400 NNs and 1024 inducing points as in the paper). We will add similar results to the paper to reassure readers on this point.
>
> 5. We agree that the rate of convergence is of practical value and we have preliminary results for this but wish to delay publication of them (see 5.1). Nevertheless we believe the importance of sharing existing results with the community outweighs delaying publication for completeness. This view appears to be supported by the other reviewers.
>
> 6. GPnn _is_ designed for interpolation tasks on which we demonstrate excellent performance, even in the presence of somewhat severe misspecification, at modest expense. It is ongoing work to generalize our existing results and determine the dependence of the rate on the kernel properties.
> Practitioners generally fix a choice of kernel(s) and then optimize hyperparameters in relation to that choice. A key point that our paper makes is to caution against excessive effort in hyperparameter fine-tuning. Our empirical and theoretical results demonstrate that the predictive measures commonly used by the community are robust to non-optimal parameter specification for pre-picked kernels, which runs somewhat counter to conventional belief.
>
> 7. As covered under bullet point 2, improving GP regression is of major importance in its own right so we did not view a failure to address non-regression applications, e.g. classification, to be a weakness of the paper. In fact, we note that the use of GP for regression tasks is more common than for classification, and thus we would argue that our findings, although focussed in the regression domain, have wide reaching and significant impact to the community. Having said that, we take on board your comments on the potential advantages of variational methods beyond regression whilst also remaining open minded as to whether GPnn could potentially play a role in that domain.
>
> 8.
> - Re "Algorithm 11 seems flawed": Thank you for pointing out the need to justify this. We are confident the simulation is both valid and appropriate for what we are aiming to do and hope the following brief explanation helps: Firstly, we are focussing on point prediction and lack of correlation between test points does not impact that. Secondly, the validity of the estimates is provable: The gist of the argument is to show for fixed size-$n$ training set $X$ that $\\{(x_i^*,N(x_i^*),y_i^*,y_i)\\}\_{i=1}^{n^*}$ is a set of iid vector-RVs (here the $y_i$ are m-dimensional). Then since $e_i^*$ is a deterministic function of $(x_i^*,N(x_i^*),y_i^*,y_i)$, $\\{e_i\\}\_{i=1}^{n^*}$ is also a set of iid RVs. Finally $e^* =  E(e_i)$ so that $e^* = E[\frac{1}{n^*} \sum_{i=1}^{n^*} e_i]$ as required. The same argument holds for $l^*$ and $z^*$.
> - Re "gives GPnn an advantage": We do not believe this to be the case because (a) the estimated mean mse, nll and calibration are valid for (point-estimate) GPnn and (b) we are in any case only using this simulation to experimentally demonstrate convergence toward the theoretical limits of theorem 1, not to compare GPnn performance with the other methods.

---

> > ### Comment · Reviewer_N8D3 · 2023-08-15
> > **Post-rebuttal comments**
> >
> > I appreciate the authors for their rebuttal and for other reviewers for their insights, and maintain the same assessment as before with a reduced confidence score.
> >
> > I have gathered more insight into the the interplay between GP models and its scalable kNN variants from the rebuttal and subsequent discussion from other reviewers. The paper “Hierarchical Nearest-Neighbor Gaussian Process Models for Large Geostatistical Datasets” mentioned by reviewer 3iYQ is an important precursor work and bridges this paper with the original full GP model: we do not need to see nearest-neighbor GPs as a straightforward approximation to GP models, but as a hierarchical model with a nearest-neighbor element in its own right. I believe that this paper gives us more insight in how to bridge hierarchical NN-GP with the GP model that uses all the training data points. I have, however, found that the theoretical results presented in the paper to have a quite limited impact on how we understand this connection.
> >
> > The paper considers the $n\rightarrow\infty$ limit behavior for virtually all its theoretical findings, but it is the infinite training data regime that yields many confounding results: when we are presented with infinitely many training data, the $m$ nearest neighbors of an arbitrary test point is itself (or a selection of points within an arbitrarily small ball surrounding the test point). In this asymptotic scenario, no kernel hyperparameter would ever matter as the kernel matrix converges to a matrix of all $1$s, hence the result negating the purpose of kernel learning from the asymptotic sense. This also shows why evaluating the variance parameter in the observation model still matters: the nearest neighbor observed value is essentially a collection of observation centered around the ground truth value with variance $\sigma_\xi^2i$. The paper presents a counterintuitive result in Remark 2 that isotropic kernels already suffice for best MSE in the asymptotic sense, and I argue that this line of counterintuitive reasoning could go one step further as we are in the domain of infinite data: kernels with infinitely small length scale is the optimal choice for all regression tasks, as it offers the best flexibility in the prior space and all the rest of Theorem 1 still holds. I unpack this line of reasoning in order to show that the infinite limiting behavior might tell very little about this model in a realistic setting, to an extent bordering on no realistic meaning. I am not, however, using this logic to discount, for example, the empirical convergence result presented in Figure 2.
> >
> > The author demonstrate in the rebuttal that they think evaluating GPnn using Algorithm 1 is fair practice, but I am still unsure whether it is true, and I phrase my question in a more straightforward manner. I assume training and test data should be generated as an entire set at first in one go, and then partitioned, but Algorithm 1 treats training and test data differently. Could you tell me why generating test data conditioned only on its nearest neighbors does not make it easier to predict based on its nearest neighbors?
> >
> > I agree after some re-consideration that applying GP models along with kNN carries some utility in certain applied domains (for example, geostatistics), and this is the main reason for the reduction in my confidence score. I remain unconvinced about this paper’s theoretical results, as I believe that a true leap in understanding this type of models would involve some degree of non-asymptotic analysis.

---

> > > ### Author Response · Authors · 2023-08-17
> > >
> > > Thank you for your additional comments. We take on board and to some extent agree with your view on the importance of non-asymptotic results. We hope to produce such results that explicitly depend on the kernel parameters in the near future. Similarly, we understand your reservations and example regarding "infinitely small lengthscales" but argue that even without explicit convergence rate results, empirical results confirm that this approach (in less extreme circumstances) has very good performance and exhibits convergent behaviour, despite obviously existing in the finite data regime.
> > > In addition, we would point out that although rates obviously depend on the choice of kernel, as an example if we consider an isotropic kernel we can infer that the corresponding convergence will depend on the convergence of the distance between neighbouring input points, something which has been studied extensively in the classical kNN literature and shown to display gradual convergence rates (e.g. [1],[2],[3]). As such, we expect similarly gradual convergence for a large number of commonly-used kernels.
> > >
> > > Thank you for your insightful feedback and clarification regarding Algorithm 1. We now further understand your concerns, but remain confident in the validity of our conclusions pertaining to convergence behaviour. We believe that we can probe your concern by supplementing the current version with a computationally cheap deterministic function. Having applied this enhancement already to Figure 2 using the Oakley and O'Hagan function from Section 7.2, we can report that the plots maintain their core characteristics consistent with the original figure, while directly addressing the issues you raised. We can also run (smaller scale) versions to generate very similar plots to Figure 2 using data sampled directly from a GP and avoiding the use of Algorithm 1, further reinforcing our confidence in our conclusions.
> > >
> > > We hope that our successful reproduction of the salient features of figure 2 via alternative means will have already satisfied you on the matter of Alg. 1. However, we realise that the above does not fully explain its validity in response to your final question on the topic. Details are given below. If we include algorithm 1, in addition to other evidence, we will include a proof of its validity in the appendix.
> > >
> > > Define "Algorithm 1" as in the paper. As argued in our original rebuttal, the evaluations ${e_i^*}$ are iid RVs and so our estimator, $\frac{1}{n^*}\sum_{i=1}^{n^*} {e_i^*}$ is valid for the corresponding expectation.
> > >
> > > Define "Algorithm 1b" to be the procedure whereby we generate an $n$-length $x$ training set that we subsequently hold constant. For each generated test point $x^*$ we then generate an $(n+1)$-length GP sample $y$. We take the $m$ nearest-neighbours of the test point $x^*$ and evaluate the function $e(\cdot)$ to obtain ${e_i^*}'$. We repeat this $n^*$ times and compute a Monte-Carlo estimate of the expectation, $\frac{1}{n^*}\sum_{i=1}^{n^*}{e_i^*}'$. This method is clearly valid, albeit computationally very expensive.
> > >
> > > Finally, we show that the expectations in both cases are equivalent, where we begin with the case of Algorithm 1b and show that it is equivalent to that of Algorithm 1.
> > >
> > > We use $y^*$ to refer to the test observation, $y'$ to refer to the nearest-neighbours and $y''$ to the remaining disjoint observations. $y=(y^*,y',y'')$ refers to the full (n+1) length vector and hence $p(y)$ to the full joint distribution:
> > >
> > > $$
> > > 	\frac{1}{n^*}\sum_{i=1}^{n^*} {e_i^*}' \approx E_{(y^*,y',y'')}[e(y^*,y',y'')]
> > > 	 = E_{(y^*,y',y'')}[e(y^*,y')]
> > > 	= E_{(y^*,y')}[e(y^*,y')]
> > > 	\approx \frac{1}{n^*}\sum_{i=1}^{n^*} {e_i^*}
> > > $$
> > > Since $e(\cdot)$ is only a function of $y^*,y'$.
> > >
> > > [1] L ́aszl ́o Gy ̈orfi et al. A Distribution-Free Theory of Nonparametric Re- gression. 2010.,
> > >
> > > [2] Kohler, Krzyzak, and Walk, ‘Rates of Convergence for Partitioning and Nearest Neighbor Regression Estimates with Unbounded Data’,
> > >
> > > [3] Kulkarni and Posner, ‘Rates of Convergence of Nearest Neighbor Estimation Under Arbitrary Sampling’.

---

> > > > ### Author Response · Authors · 2023-08-21
> > > > **Edit**
> > > >
> > > > Please note that we have both amended and amalgamated our previous discussion comments via the editing process.

---

### Official Review · Reviewer_MqBs · 2023-07-09

**Soundness:** 4 excellent
**Presentation:** 3 good
**Contribution:** 4 excellent
**Rating:** 8
**Confidence:** 4

**Summary:**

This paper starts by proposing use of a simple nearest neighbors scheme for prediction, where instead of using the training set, one uses the $m$ nearest neighbors of the training set to the test set. They then, in Theorem 1, analyze the asymptotic expected MSE, calibration and negative log-likelihood of this prediction approach under a given estimator of the parameters. They find that there is robustness to poor estimation except in noise variance. They show empirically that their simulations match theory in algorithm 1. They then propose a scalable GP regression algorithm: noting that when using their nearest neighbor prediction approach, the MSE, calibration and negative log-likelihood are *somewhat* robust to wrong parameters (except noise variance), they use multiple approximation steps, including using a subset of the training data and using a structured covariance matrix. In order to improve noise variance estimation, they add a calibration step. They show that it outperforms several baselines on real world datasets.

**Strengths:**

This is a very interesting paper: the resulting technique is very simple but pops out in a very non-obvious way. Further, it leads to a powerful consequence: you don't need the greatest parameter estimates for training if you do nearest neighbors prediction, as long as you add a calibration step to improve the estimate of noise variance. Despite my slight reservations about the baselines and the strange paper structure, I strongly recommend acceptance.

**Weaknesses:**

This paper is written in a very non-standard way and completely ignores the standard ML paper format. It completely ignores the standard introduction, replacing it by what is often the 2nd or third section, which is some technical background. It then has the 2nd section, which is *similar* to what often comes at the end of an intro, but is somehow different. The related work section is very short. In general everything is somewhat terse. I don't completely hate it, but it's jarring to read this, and I'm not sure what the justification for doing it this way is, particularly since the authors seem aware of literature and the techniques used in the literature so presumably are aware of how standard ML papers are written to flow a certain way.

The main weakness is that I'm not sure how close to state of the art the baselines are.

**Questions:**

Why did you use this particular, somewhat strange, paper structure?

Why is your calibration definition what it is?

I'm a little confused: you said you replace $X$, but eqn. 5 still has it. Is this a typo?

Is SVGP really state of the art? What about SKI [1] and its extensions?

[1] Wilson, Andrew, and Hannes Nickisch. "Kernel interpolation for scalable structured Gaussian processes (KISS-GP)." International conference on machine learning. PMLR, 2015.

**Limitations:**

Yes.

---

> ### Author Rebuttal · Authors · 2023-08-08
>
> 1. Thank you for your comments on the structure. It was not a conscious decision to write substantially outside the stylistic norms of the community and on reflection perhaps we should have tried to conform more strongly. We do not wish to drastically change the structure of the paper at this stage in case that would lead to a requirement for other reviewers (who did not object) to reassess. We hope that you do not think this unreasonable and will definitely take your comments on board for future work. We are planning to elaborate and extend the related work section.
>
> 2. We define calibration this way for a few reasons:
> 	- For a well-specified GP, the $E_Y[MSE]$ matches the predictive variance, i.e. the uncertainty in mean prediction (or the magnitude of the residuals) is reflected in the variance predicted by the model.
> 	- It provides a convenient numerical baseline for what "well-calibrated" means, when the value is 1.
> 	- It allows us to carry out our simple recalibration procedure (Algorithm 2), to improve both our measure of calibration and NLL, whilst leaving MSE unchanged.
> 	- Although a "weak" measure of calibration, obtaining a  value near 1 is a necessary condition for effective calibration, and marked departures from 1 were detected for some of the methods in figure 4 and table 3.
> 		Additionally, we have subsequently found that [1] uses the same definition and we will reference that in the paper.
> [1] Jankowiak, Pleiss, and Gardner, ‘Parametric Gaussian Process Regressors’.
>
> 3. This is not a typo, but we appreciate your pointing this notational ambiguity out so that we can rectify this. In a sense, under this model specifying $X$ and $N(x^*)$ are equivalent, since $N(x^*)$ is directly derived from the training set $X$.
>
> 4. Given the potential for impact that the findings in this paper imply, we were keen to ensure that our benchmarks included widely used methods, so that an interested user might easily determine the potential for these findings to inform their work.  We are not aware of a method which has replaced SVGP in this position, in terms of generality of uptake and usage. We were aware of the paper on SKI that you cited, but were under the impression it had limited applicability to datasets with more than a handful of dimensions. We would like to thank you for exposing us to the various extensions however, and would aim to include them as additional baselines to compare against in future publications.  We do not believe that the omission of this comparison harms the conclusions which we draw in this paper, however.

---

> > ### Comment · Reviewer_MqBs · 2023-08-15
> >
> > Thank you for your response. You are right that SKI suffers from the curse of dimensionality. The extensions (e.g. [1]) have not yet taken the place of SVGP, although you should probably mention them and (if you have time) add a comparison of one of them in the final version. My score remains unchanged.
> >
> > [1] Yadav, Mohit, Daniel R. Sheldon, and Cameron Musco. "Kernel Interpolation with Sparse Grids." Advances in Neural Information Processing Systems 35 (2022): 22883-22894.

---

> > > ### Author Response · Authors · 2023-08-15
> > >
> > > Thank for your further comments. We will follow up on the suggestions you have made and will endeavour to make comparisons in this  paper, and certainly add SKI-based methods to future evaluations.

---

### Official Review · Reviewer_jFQi · 2023-07-12

**Soundness:** 3 good
**Presentation:** 3 good
**Contribution:** 3 good
**Rating:** 7
**Confidence:** 4

**Summary:**

 This work at one level is about proposing a scalable GP approach called GPnn where the prediction step uses only the nearest neighbours of a test point in order to form the predictive distribution. This approach is well-motivated as the authors argue that there isn't a strong mathematical reason to couple the training step / estimation of hyperparameters and the prediction / generalisation over unseen inputs. At the end of the day, we care about the latter (prediction performance) and not really the former. At another level, there is a theoretical component where they show that predictive distributions obtained through GPnn are robust to a wide array of model misspecification, for instance, wrong kernel choice or Gaussian noise in the large data limit.

**Strengths:**

The paper is well-written and clear. There is always a demand for new GP approximations which scale to millions of data points as SVGP has a known issue with over-estimating the aleatoric uncertainty. The algorithms clearly describe the simulation and evaluation step. There is a good summary of other distributed methodologies and clear delineation of how this work is different. Some components from this paper can be applied to other methods in order to refine estimates and calibration (Remark 5).



**Weaknesses:**

Figure 2 is hard to read - there is a lot going on and ideally should have been plotted with N on the x-axis.

**Questions:**


- What about a sensitivity analysis to the choice of M? or are GPnn predictions insensitive to this in the large data limit.
- Can M be learnt instead of set?
- Perhaps some areas of the input space require more M for good predictive performance than areas where the function isn't changing much.
- It would be interesting to see the performance on non-stationary functions.
- There isn't sufficient insight on the selection of nearest neighbours  exact or approximate - at first blush one would dismiss the idea because of the added compute entailed in finding neighbours per test data point. What is the complexity of this as a function of N and d precisely?

**Limitations:**

There is some discussion of limitations.

---

> ### Author Rebuttal · Authors · 2023-08-08
>
> 0. We agree that the figure is not the most straightforward to interpret, but since the intention was to convey the dependence of the performance metrics on the estimated parameters, for varying dataset sizes, reproducing the figures with n along the x-axis would lose much of this information (please see the pdf attachment for our best attempt). For example the third plot in the paper shows the  tendency, with increasing n, for "MSE verses $l$" to flatten toward the limiting horizontal line  (i.e. for MSE to become insensitive to $l$ ). For the final version of the paper we will modify our caption and text to clarify, and may include an additional figure with n along the x-axis, although we suspect the information this conveys will be largely covered by Fig. 5 (which will now reference in 5.2).
>
> 1. Theorem 1 in the paper indicates the degree of sensitivity of prediction performance to m in the large n limit, e.g. for MSE, noise variance scaled by 1/m. For finite n it might be interesting to do such an analysis. We chose the value of m in our experiments without much experimentation beyond a brief comparison on otherwise unused synthetic data. We found minimal impact on performance. For even larger datasets it may be possible to reduce m and decrease training and test time at little cost to performance, but we did not investigate this further since we wished to emphasis the minimal amount of tuning our method requires.
>
> 2. Yes, for example by cross-validation, as is commonly done in the literature (on kNN). This would of course add additional computational cost, but could be done efficiently by starting with a maximum number, and iteratively evaluating performance on decreasing subsets.
>
> 3. This is an interesting but non-trivial suggestion which we would like to explore in future refinements to the method.
>
> 4. Our current theoretical analysis relies on the stationarity of both the generative and predictive model, although we suspect this could be relaxed. We appreciate the potential interest in non-stationary functions, but do note that results using the Oakley and O’Hagan function (non-stationary) are included in 7.2 and provide initial encouragement.
>
> 5. The literature on wide ranging approximate kNN methods is extensive and it is difficult to find a general complexity for the procedure. We used the SciKit-Learn implementation, whose costs are described in the associated documentation, e.g. O(dlogn) query compute-cost for the Ball-tree algorithm which the default automated algorithm selection in SciKit-Learn should at least match. In contrast, exact kNN costs O(dn) which is why we have chosen approximate kNN in preference. In practice the predictive timings are fast enough for most applications, taking on a laptop, for example, approximately 0.06s per prediction on the high-dimensional (d=378 and therefore relatively extreme) Ctslice dataset. We will add both example timings and the N,d complexities to the paper, which should also reassure readers that the cost of kNN search is not detrimental to the test-times of GPnn in practice (see also our response to reviewer N8D3 bullet point 4 and table 1 from the pdf).

---

> > ### Comment · Reviewer_jFQi · 2023-08-14
> > **Post rebuttal**
> >
> > Thank you for responding to the questions and I would encourage you to action point 0 and add details on 5 in the final manuscript.
> >
> > Otherwise, I am happy to persist my score.

---

> > > ### Author Response · Authors · 2023-08-15
> > >
> > > Thank you for your feedback, we will follow your recommendations.

---

### Author Rebuttal · Authors · 2023-08-08

We thank all of the reviewers for their useful and constructive feedback and for the time and effort that they have voluntarily set aside for this task.

---

### Decision · Program_Chairs · 2023-09-21

**Decision:**

Accept (poster)

**Comment:**

This paper investigate nearest neighbor approximations to GP posteriors and provides a theoretical asymptotic analysis. The proposed method offers many advantages over SVGP (a scalable GP approximation used extensively by the ML community), which the authors demonstrate experimentally. Moreover, despite its asymptotic limitations, the theoretical analysis is sounds and provides interesting insights. Based on these strengths, I recommend this paper is accepted into NeurIPS.

However, there is a glaring weakness off this paper, which I would strongly encourage the authors to address in the revisions: the treatment of prior NN-approximation work from the GP literature. While the authors nod to a few recent works, they do not include any discussion of them, and how these approaches differ from what is proposed. Most notably, the authors have also missed the pioneering work of Vecchia (1988) and many of the follow-up works - a method which has much popularity in the geostatistics community. The theoretical analysis of the authors is an important contribution; however, I fear that this analysis might not actually be too useful since the NN approximation analyzed by the authors differs significantly from these well-established/highly used approximations.

I would encourage the authors to:
- Include more relevant (and popular) NN-based GP approximations, on top of the more recent ones in the related work section
- Provide a substantive description of these methods and how the proposed method differs. (Would the same theoretical analysis apply to these other methods?)
- Include an experimental comparison of the proposed method against other NN-based methods.

Vecchia, A. (1988). Estimation and model identification for continuous spatial processes. Journal of the Royal Statistical Society, Series B, 50(2):297–312.